

# Spatio-seasonal variability of chromophoric dissolved organic matter absorption and responses to photobleaching in a large shallow temperate lake

María Encina Aullo-Maestro[1], Peter Hunter[1], Evangelos Spyrakos[1], Pierre Mercatoris[1,2], Attila Kovács[3], Hajnalka Horváth[3], Tom Preston[4], Mátyás Présing[3], Jesús Torres Palenzuela[5], Andrew Tyler[1]

[1]Biological and Environmental Science, School of Natural Sciences, University of Stirling, Stirling, UK

[2]Plymouth Marine Laboratory, Plymouth, UK

[3]Balaton Limnological Institute, MTA Centre for Ecological Research, Tihany, Hungary

[4]Stable Isotope Biochemistry Laboratory, SUERC, East Kilbride, UK

[5]Remote Sensing and GIS Laboratory, Department of Applied Physics, Sciences Faculty, University of Vigo, Vigo. Spain

*Correspondence to*: M.E.Aullo-Maestro (meaullo@gmail.com)

**Keywords.** Chromophoric dissolved organic matter (CDOM); Photobleaching; Bio-optical properties; Spectral absorption; Lake Balaton.



**Abstract**

The development and validation of remote sensing-based approaches for the retrieval of CDOM concentrations

requires a comprehensive understanding of the sources and magnitude of variability in the optical properties of

dissolved material within lakes. In this study, spatial and seasonal variability in concentration and composition of

CDOM and the origin of its variation was studied in Lake Balaton (Hungary), a large temperate shallow lake in

central Europe.  In addition, we investigated the effect of photobleaching on the optical properties of CDOM

through in-lake incubation experiments. There was marked variability throughout the year in CDOM absorption

in Lake Balaton ($a_{CDOM}$ (440) = 0.06 - 9.01 m$^{-1}$). The highest values were consistently observed at the mouth of

the main inflow (River Zala), which drains humic-rich material from the adjoining Kis-Balaton wetland, but

CDOM absorption decreased rapidly towards the east where it was consistently lower and less variable than in

the westernmost lake basins. The spectral slope parameter for the interval of 350–500 nm ($S_{CDOM}$(350–500)) was

more variable with increasing distance from the inflow (observed range 0.0161-0.0181 nm$^{-1}$ for the mouth of the

main inflow and 0.0158-0.0300 nm$^{-1}$ for waters closer to the outflow). However, spatial variation in $S_{CDOM}$ was

more constant exhibiting a negative correlation with $a_{CDOM}$(440). DOC was strongly positively correlated with

$a_{CDOM}$(440) and followed a similar seasonal trend but it demonstrated more variability than either $a_{CDOM}$ or $S_{CDOM}$

with distance through the system. Photobleaching resulting from 7 days exposure to natural solar UV radiation

resulted in a marked decrease in allochthonous CDOM absorption (7.04 to 3.36 m$^{-1}$, 42% decrease) and an even

greater decrease in the absorption of autochthonous CDOM (1.34 to 0.312 m$^{-1}$, 77% decrease). Photodegradation

also resulted in an increase in the spectral slope coefficient of dissolved material. Terrestrial subsidies of dissolved

organic matter are known to exert a profound influence over the biogeochemistry and metabolism of lakes.  The

results from this study show that localized inputs of dissolved matter from wetlands can exert a strong influence

over the spatial and seasonal dynamics of CDOM in lakes.



## 1. Introduction

### 1.1. Importance of CDOM in lakes

There are approximately 117 million lakes on Earth greater than $0.002 \ km^2$ in surface area collectively covering about 3.7 % of its non-glacial surface (Verpoorter et al. 2014). The importance of the role that lakes play as

regulators of the carbon cycle and thereby global climate has only recently been recognized (Tranvik et al. 2009), acting as both a sink (sediment storage through flocculation from dissolved to particulate organic carbon) or source for carbon (degradation and resulting mineralization to $CH_4$, $CO$ and $CO_2$; Cole et al. 2007; Benoy et al. 2007; Tranvik et al. 2009). As a result they also play an important role in transforming and releasing terrestrially-derived carbon to the atmosphere and ocean (Tranvik et al. 2009). As extremely sensitive ecosystems (Millennium

Ecosystem Assessment 2005; IPCC 2007), lakes can respond rapidly to external pressures including meteorology, climate and land use change. This has led to the emerging concept of lakes as sentinels of environmental change (Adrian et al. 2009; Williamson et al. 2009a; Schindler 2009; Williamson et al. 2009b).

The optical properties of lakes provide particularly useful metrics for measuring ecosystem change (Vincent et al. 1998) as they not only convey information on the quantity of particulate and dissolved material but also its quality

(Williamson et al. 2014). Furthermore, understanding how the optical properties of particulate and dissolved material in lakes influences the underwater light field and water-leaving radiative signal is important for the development and application of remote sensing techniques for lake monitoring and assessment, but also their application to lake carbon studies.

Much of the dissolved organic matter (DOM) found in lakes typically represents between 90 to 100 % of the total

carbon pool (Wilkinson et al. 2013) and is derived from terrestrial inputs, transported through streams, rivers and wetlands. This allochthonous component of the DOM originates from soils, sediments and plants and is primarily composed of humic substances. The autochthonous fraction of DOM is produced mostly by phytoplankton, zooplankton and bacterioplankton and is largely composed of fulvic acids, carbohydrates, amino acids, proteins, lipids and organic acids.

Chromophoric dissolved organic matter (CDOM) is the coloured fraction of DOM. It is one of the dominant colour-forming constituents in lakes: it not only exerts a strong influence over the underwater light field and water-leaving radiance but it also has a number of important ecosystem functions. First of all, it absorbs light strongly in the ultraviolet (UV) spectrum limiting the penetration of biologically-damaging UV-B radiation providing protection for phytoplankton and other primary producers (Hoge et al. 1995; Laurion et al. 2000; Zhang et al.



2007; Williamson et al. 2001). In addition, CDOM can act as a source of nutrients whilst also of importance for

phytoplankton nutrition, and this fulfilling important role in the development of phytoplankton blooms and lake

metabolism more widely. On the other hand, studies have also shown that light absorption by CDOM can reduce

the amount and quality of photosynthetically active radiation (PAR) available to phytoplankton, thereby

decreasing primary production and constraining lake metabolism (Kirk 1994; Laurion et al. 1997, 2000; Vähätalo

et al. 2005). Moreover, its conservative behaviour with dissolved organic carbon (DOC), means CDOM is often

used as a proxy for DOC. Thus, there is substantial interest in the use of CDOM as an optical tracer of DOC due

to the importance of the latter in regulating physical, chemical and biological properties of lakes. It is therefore

important that we develop a better understanding of the optical properties of CDOM and how these relate to the

chemical composition and concentration of DOM whether driven by changes to source relationships or through

the in-lake processes and transformation of the carbon pool.

Understanding how the optical properties of CDOM vary both temporally and spatially within lakes and how

the observed variability influences the underwater light field is of particular importance for the development and

validation of remote sensing-based approaches for retrieving CDOM concentrations. The recent launch of new

platforms such as Sentinel-2 and -3, allied with the prospect of new hyperspectral sensors (e.g., EnMAP), has

provided a new impetus for the development and application of remote sensing techniques for the assessment and

monitoring of inland water quality. However, CDOM is the arguably most challenging water quality parameter

for reliably estimation from remotely sensed observations (Palmer et al. 2015) and, in spite of its importance to

the physical, chemical and biological function of lakes, it remains one of the least studied parameters. Indeed, few

studies have explored application of remote sensing for the estimation of CDOM in lakes. To progress such

research, an improved understanding of the spatial and temporal variation in the optical properties of CDOM in

lakes is needed.

### 1.2. Optical properties of CDOM

CDOM concentration is commonly measured by its absorption coefficient ($a_{CDOM}$) at 440 nm, whereas its structure

and composition has been most commonly inferred from the spectral slope parameter ($S_{CDOM}$) calculated between

two reference wavelengths (Helms et al. 2008; Fichot & Benner 2012). Other optical metrics related to CDOM

compositions include the E2:E3 ratio or M value, which is the ratio of absorption at 250 nm and 365 nm. De Haan

and De Boer (1987) used E2:E3 ratio to track changes in relative size on CDOM molecules: increases in molecular



size result in decreases in the E2:E3 ratio because of stronger light absorption by higher molecular weight compounds at longer wavelengths.

In addition, Weishaar et al. (2003) introduced the specific UV absorbance parameter (SUVA$_{254}$) defined as the UV absorbance at 254 nm normalised by the dissolved organic carbon (DOC) concentration. SUVA$_{254}$ has been shown to be strongly correlated with DOM aromaticity in a large number of aquatic environments, with higher SUVA$_{254}$ values indicative of a higher abundance of aromatic compounds. Previous studies have used SUVA$_{254}$ to explore variability in the composition of DOM in natural waters.

The compositional properties of CDOM vary over time in response to processes such as microbial decomposition and exposure to UV irradiation. Previous studies have shown the latter process, first described by Wipple (1914) as 'photobleaching', plays a major role in the transformation of DOM in natural waters. Exposure to solar irradiance has been shown to reduce its capacity to absorb light, the loss of absorptivity is linked to a reduction in molecular weight (MW), alteration of its chemical composition and an increase in the bioavailability of DOM

(Geller 1986; Keiber et al. 1990; Wetzel et al. 1995; Lindell et al. 1995; Corin et al. 1996; Reche et al. 1998) with implications for lake metabolism. $S_{CDOM}$ is also known to vary in response to photobleaching (Swan et al. 2012; Fichot & Benner 2012).

Most previous studies on the origin, distribution and degradation of DOM and how this influences the optical properties of CDOM have been undertaken in oceans (Andrew et al. 2013; Matsuoka et al. 2014; Hancke et al.

2014; D'Sa et al. 2014), coastal waters (Stedmon et al. 2000; Vantrepotte et al. 2007; Kutser et al. 2009; Para et al. 2013) or in a few cases high latitude lakes (Ficek et al. 2011; Ylöstalo et al. 2014). Understandably, the bias towards high latitude systems partly reflects the fact this region contains a high density of humic-rich lakes. In comparison our understanding of the spatial and seasonal variation in CDOM optical properties in temperate lakes at lower latitudes is comparatively poor. In these lakes, although the DOC pool is typically smaller than at higher

latitudes the influence of processes such as photobleaching are likely to be more pronounced.

In this study we explore spatial and seasonal variability in optical properties of CDOM in Lake Balaton, a large temperate lake with a highly continental climate. We investigate how changes in spectral absorption, spectral slope coefficients, SUVA$_{254}$ and the E2:E3 ratio can be used to infer information on the concentration, source and decomposition of CDOM. The main objectives of the study were to: (1) characterize the spatial and seasonal

trends in CDOM in Lake Balaton; (2) determine the origin and magnitude of the variability of different sources





of CDOM; and (3) consider the implications of variability in the concentration and composition on the underwater light field.

## 2.  Material and methods

### 2.1.  Study site

With a surface area of 596 km$^2$ and a mean depth of 3.25 m, Lake Balaton in Hungary is the largest shallow lake in central Europe (Fig. 1a).  The region is situated on the boundaries between the Mediterranean, continental, and oceanic climatic zones, resulting in a climate characterised by dry summers and moderately wet winters with typical continental extremes in temperature. The first two weeks of January are the coldest periods of the year (-4 – 3 °C) whilst July and August the warmest months (15 – 28 °C). The annual precipitation in the Lake Balaton

region is between 500–750 mm; most precipitation falls during the spring, while the minimum occurs during the summer. There is a secondary maximum in autumn, due to a strong cyclone activity at this time of the year. In regards to solar radiation, Lake Balaton is situated between the southern, western and central Transdanubian regions in Hungary with an annual mean of 4500 MJ m$^{-2}$. The highest solar radiation is received in July (650 MJ m$^{-2}$), while cloudy weather and shorter days mean that lowest radiation occurs in December. The maximum in

sunshine duration is also reached in July with more than 250 hours, falling to a minimum of approximately 40 hours during winter months.

Lake Balaton is usually divided into four basins (south-west to north-east): Keszthely; Szigliget; Szemes; and Siófok (Fig. 1b).  The lake has 20 permanent and 31 temporary inflows, many of which are small streams or springs in the lake bed. The largest inflow to the lake is the Zala River, which flows through Kis-Balaton reservoir

– a large semi-natural wetland system – and enters the lake in the westernmost part of Keszthely basin (Fig. 1b). The only outflow is the Sió channel in the northeast that connects the Siófok basin with the Danube River.

Lake Balaton has experienced eutrophication since the middle of the 18[th] century due to agricultural intensification and urbanisation within the catchment. Since the early 1980s, significant effort has been invested in improving its water quality (Tátrai et al. 2000). The construction of Kis-Balaton reservoir and wetland system was one of the

main engineering controls built to reduce nutrient inflow from the Zala River and the overall loading within the lake. Kis-Balaton removes approximately 60 % of the annual nutrient loading to Lake Balaton (Szilágyi et al. 1990).  However, nutrient inputs from the Zala River still result in high summer primary production in the eutrophic (>20 mg m$^{-3}$ chl-a) waters in the western basins, with a steep gradient towards more mesotrophic (2 - 20 mg m$^{-3}$ chl-a) waters in the east. The hypertrophic Kis-Balaton wetland system is believed to be responsible



for much of the DOM entering the lake, largely derived from luxuriant growth and decomposition of aquatic plants. Previous research (Palmer et al. 2013; Riddick et al. 2015) has shown that CDOM is usually significantly higher close to Zala River inflow, and decreases towards the outflow but very little is known about the seasonal dynamics of CDOM in the system.

Suspended particulate matter in Lake Balaton is highly variable (spatially and temporally) due to its very shallow
depth, constant mixing and susceptibility to wind-driven resuspension events (Istvánovics et al. 2004). Phytoplankton composition in the lake shows strong seasonal trends, with two annual blooms (Padisak & Reynolds 1998; Présing et al. 2008; Hajnal & Padisák 2008). In late summer and early autumn, cyanobacterial blooms often occur in the Keszthely basin (I), extending westwards to the Szigliget (II) and Szemes (III) basins and very occasionally to the Siófok (IV) basin. The lowest phytoplankton biomass generally occurs in February
when the lake can be ice-covered; a less noticeable dinoflagellate bloom may also occur in April (Mózes et al. 2006).

### 2.2. Water sampling

Spatial variability in CDOM quantity and quality was assessed over a 1-week period in July 2013 (6 stations) and a 3-week period in July 2014 (25 stations) at 31 stations over a biogeochemical gradient from the southwest in the
water masses influenced by Zala River to the northeast near the outflow (Fig. 1c). Five stations were also sampled in the Kis-Balaton reservoir during the same period (2 in 2013 and 3 2014). These intensive sampling campaigns were timed to coincide with the annual summer peak in DOC to capture the maximum spatial variability likely to occur in the system.

In order to capture seasonal variability in CDOM quantity and quality, water samples were collected fortnightly
at 6 long-term monitoring stations on Lake Balaton. These comprised stations 01 and 03 from Keszthely basin (I), station 12 from Szigliget basin (II), station 20 from Szemes basin (III) and stations 25 and 30 from Siófok basin (IV) (for location of stations see Fig. 1c).

Water samples for DOC analysis were collected in triplicate using acid-rinsed polypropylene bottles at 0.3 m depth below the surface. The samples were immediately stored on ice and in the dark until they were transferred
to the laboratory for filtration. The samples were filtered through 0.7 µm pre-combusted 47 mm glass-fibre membranes (Whatman GF/F) and stored cold (4°C) and in the dark until measurement. Samples for CDOM analysis were collected separately in acid-rinsed amber glass bottles from 0.3 m depth and immediately stored on ice and in the dark until transfer to the laboratory. Samples were pre-filtered through pre-combusted 0.7 µm pore size glass-fibre membranes (Whatman GF/F) to remove large particles and then re-filtered through a 0.2 µm



Whatman nucleopore membrane filters. The samples collected as part of the seasonal sampling campaign were

measured fresh (i.e., without preservation) within 24 hours following Tilstone et al. (2002). The samples collected

during the intensive summer campaigns were preserved with a 0.5 % (vol:vol) solution of 10 g $L^{-1}$ of sodium azide

(NaN$_3$) (Ferrari et al. 1996) prior to analysis, which was completed within 1 month of sample collection.

### 2.3.    CDOM absorption

The spectral absorbance ($A$) of the seasonal samples was measured on a Shimadzu UV 1601 spectrophotometer

(Cuthbert & Del Giorgio 1992) using a 1, 4 or 10 cm cuvette between 350 and 800 nm with a 0.5 nm sampling

interval using ultrapure water (Milli-Q) as a reference (Vodacek et al. 1997). Samples from the intensive

campaigns were measured on a Cary-100 UV-visible spectrophotometer using a 1 or 10 cm cuvette between 200

and 800 nm with 0.2 nm sampling interval using ultrapure water with the addition of 0.5 % (vol:vol) sodium azide

as the reference. The absorbance data were baseline corrected by subtracting the mean of $a_{CDOM}$ in a 5 nm interval

centred at 685 nm (after Babin et al. 2003). This wavelength was selected because absorption by CDOM and other

dissolved constituents is negligible in the far red (Pegau et al. 1997).

The CDOM spectral absorption coefficient ($a_{CDOM}$) was calculated as follows (Kirk 2010):

$$a_{CDOM}(\lambda, \mathrm{m}^{-1}) = 2.303 \cdot \frac{A_{CDOM}(\lambda)}{L} \tag{1}$$

where $a_{CDOM}(\lambda)$ is the absorbance over a pathlength of $L$ meters.

We used the CDOM absorption coefficient at 440 nm for describing changes in CDOM quantity. The spectral

slope for the interval of 350–500 nm ($S_{CDOM}(350–500)$) (Babin et al. 2003) was determined by fitting a single

decreasing exponential function to the absorption spectra using non-linear regression (Bricaud et al. 1981;

Twardowski et al. 2004) between 350 and 500 nm, as follows:

$$a_\lambda(\mathrm{nm}) = a_{\lambda_{ref}} \cdot e^{-S(\lambda - \lambda_{ref})} \tag{2}$$

where $\lambda_{ref}$ is a reference wavelength (440 nm in this study).

The E2/E3 index was calculated as the ratio of the CDOM absorption coefficients at 250 and 365 nm. Previous

studies have shown that decreases in this ratio are related to increases in molecular size, aromaticity and

humification of DOC (Peuravuori & Pihlaja 1997).  Finally, specific UV absorptivity at 254 nm (SUVA$_{254}$) was

obtained by normalising the absorption at 254 nm by the DOC concentration (mg $L^{-1}$) (Weishaar et al. 2003).

Because aromatic groups are predominantly responsible for absorption at this wavelength, the index indicates the

degree of DOC aromaticity as shown in Helms et al. (2008).




### 2.4. Dissolved organic carbon (DOC)

Samples for dissolved organic carbon (DOC) were measured by thermal catalysis at 950°C in an Elementary High

TOC instrument (Elementar Analysensysteme GmbH Germany) equipped with a platinum catalyst cartridge using

synthetic air as the carrier gas.

### 2.5. CDOM photodegradation

In order to examine the effects of solar radiation on autochthonous and allochthonous CDOM in Lake Balaton, a

7-day in-lake incubation experiment was undertaken during mid-July 2014. Fifty-six CDOM samples from Lake

Balaton were incubated in 65 mL capacity quartz tubes over 7 days under natural solar radiation. The quartz tubes

were attached horizontally to a wire frame to minimise shading and submerged approximately 1 cm beneath the

water surface in a sheltered bay. Twenty-one experimental samples were composed of phytoplankton-derived

autochthonous CDOM ($CDOM_{auto}$) and a further 21 were comprised of CDOM of allochthonous origin

($CDOM_{allo}$). In addition, 14 dark control samples ($CDOM_{allo-dark}$ and $CDOM_{auto-dark}$) were incubated (7

allochthonous, 7 autochthonous).

The autochthonous CDOM was extracted from a strain of *Cylindrospermopsis raciborskii* (ACT 9502) previously

isolated from Lake Balaton and grown under nutrient replete conditions in semi-continuous culture at 24°C and

14:10 h light/dark cycle. *Cylindrospermopsis raciborskii* dominates the phytoplankton community during

summer in Lake Balaton, often contributing >90% of the total biomass. A total of 3200 mL of cultured material

was centrifuged in the early stationary growth phase (5 min, 4000 rpm; Hermle Z320 BHG) and the resulting cell

pellet was broken using a mini-bead beater (30 s, 3500 rpm; Biospec products) to facilitate the release of cell

contents. The total cell disruption was confirmed by microscopic examination (Olympus BX51). The material

was incubated in the dark for 5 days at 20°C to allow production of CDOM and then diluted to 0.4 % (vol:vol)

with ultrapure water.

For the $CDOM_{allo}$ samples, 5 L of water was collected inside the mouth of Zala River at 1 m below the surface

with an acid-rinsed amber glass bottle on day zero of the experiment. The predominately allochthonous origin of

the DOM was confirmed by mass-spectrometry (Lajtha & Michener 1994). 0.5 L of water was filtered in triplicate

and the filter was dried and stored until analysis on an Isotope Ratio Mass Spectrometer (ANCA-MS, Europa

Scientific Ltd., UK). The $\delta^{13}C$ values of the allochthonous samples analysed had a mean $\delta^{13}C$ signal of -33.48 ±

0.43, which is consistent with published data on the $\delta^{13}C$ signature of C3 plants.





The CDOM$_{auto}$ and CDOM$_{allo}$ samples were filtered through a pre-combusted 47 mm diameter glass fibre filter paper (Fisher Brand MF300, nominal pore size 0.7 μm) previously rinsed with ultrapure water to remove particulate matter including bacteria. The samples were re-filtered using 0.2 μm porosity Whatman nucleopore membrane filters. The quartz tubes were acid-washed for 24 h, and then rinsed repeatedly with ultrapure water. The tubes were then filled with the CDOM samples and sealed with parafilm to prevent contamination. The dark samples were wrapped with black vinyl tape (resistant to UV radiation). Data on the total solar UV-radiation during the experiment were obtained from the Hungarian Meteorological Service.

One CDOM$_{allo}$ and one CDOM$_{auto}$ sample were collected and analysed in triplicate at daily time steps and their absorption and fluorescence spectra were measured. CDOM absorption coefficients were measured according to the methods detailed above. Subsamples for fluorescence measurements were stored cold (4 °C) and in the dark after preservation with a 0.5 % (vol:vol) of 10 g L$^{-1}$ sodium azide (NaN$_3$) (Ferrari et al. 1996) until further analysis.

Spectral fluorescence signatures (SFS) were measured using an Instant Screener (ISC) analyser (Laser Diagnostic Instruments Ltd., Tallinn, Estonia). Measurements were made using a 1 cm quartz cuvette at excitation wavelengths from 240 to 360 nm and at emission wavelengths from 260 to 575 nm with a 5 nm slit-widths for excitation and emission wavelengths. Ultrapure water with 0.5 % NaN$_3$ was used as a reference. The fluorescence signals of the samples were examined in two spectral regions. "Protein-like" fluorescence (F$_n$(280)) was excited at a wavelength of 280 nm, with the emission peak recorded in the range 350 ± 5 nm. "Humic-like" fluorescence (F$_n$(355)) was excited at 355 nm and its emission was measured at 450 ± 5 nm (Vodacek et al. 1997; Vignudelli et al. 2004). The fluorescence data were expressed as QSU (Quinine sulfate units; Coble et al. 1998).

## 3. Results

### 3.1. Seasonal variability

#### 3.1.1. CDOM optical properties

There was marked seasonal variability in the CDOM concentration in Lake Balaton (Fig. 2). High $a_{CDOM}$ (440) values were observed throughout the year at the mouth of the Zala River in the Keszthely basin at ST01 (Table 1), with concentrations increasing from an annual minimum in spring (3.69 m$^{-1}$ in March) to a peak in August (9.01 m$^{-1}$) during the warmest and driest period of the year (Anda & Varga 2010). Values of $a_{CDOM}$ (440) decreased for ST03 (0.64 m$^{-1}$ in June – 1.43 m$^{-1}$ in March) and were consistently lower and less variable in the other lake basins with a maximum value of 0.63 m$^{-1}$ observed at ST12 (Szigliget) in September and a minimum value of 0.06 m$^{-1}$ at ST30 (Siófok) in June.





$S_{CDOM}$ (350-500) coefficients in Keszthely (I) basin varied between 0.0161 nm$^{-1}$ in August at ST01 and 0.0221 nm$^{-1}$ in June at ST03 (Fig. 2). $S_{CDOM}$ was more variable with increasing distance from the inflow; all the stations except for ST01 demonstrated a maximum in early or mid-summer month (June-July) and minima in spring and autumn (Table 1). The maximum value for $S_{CDOM}$ (350-500) was 0.0300 nm$^{-1}$ at ST30 in June and the minimum observed was 0.0158 nm$^{-1}$ at ST30 in May, highlighting the high variability of this parameter near the outflow.

$S_{CDOM}$ (350-500) values for Keszthely and Siófok basins were negatively correlated with $a_{CDOM}$ (440) (Fig. 3a & b) ($R^2$=0.78, p<0.0001 for the Keszthely basin and $R^2$=0.92, p<0.0001 for the Siófok basin). The relationship between $S_{CDOM}$ and $a_{CDOM}$ in the Szigliget and Szemes basins was also negative ($R^2$=0.91, p=0.01 for Szigliget and $R^2$=0.79, p=0.01 for Szemes) (Fig. 3b).

### 3.1.2. DOC

Seasonal variation in DOC was measured at six permanent sampling stations (Stations 01 and 03 from Basin Keszthely (I), Station 12 from Basin Szigliget (II), Station 20 from Basin Szemes (III), Stations 25 and 30 from Basin Siófok (IV). DOC concentrations ranged from 7.63 at ST25 in April to 19.70 mg L$^{-1}$ at ST01 in July with a mean value of 10.1 mg L$^{-1}$ (Table 1). The highest concentrations were observed at ST01 (where the Zala River enters the lake) in summer (July: mean 19.70 mg L$^{-1}$, August: mean 18.60 mg L$^{-1}$) and early autumn (October:

mean 18.99 mg L$^{-1}$) (Fig. 2).

Both ST01 and ST03 in the Keszthely basin exhibited two peaks in DOC, one in summer and one in autumn (Fig. 2). The summer peak was larger at ST01 near the inflow, whereas the autumn peak was larger at ST03. However, a slightly different trend was observed in the central part of Keszthely basin at ST03, where DOC started increasing in the end of the summer reaching a maximum in November (12.7 mg L$^{-1}$) whereas the maximum was

reached in July for ST01. For stations furthest from the main inflow, DOC concentrations remained relatively consistent during the course of the year, with two small peaks at ST12 in July and November. DOC concentrations at ST20, 25 and 30 were significantly lower than ST01, 03 and 12.

### 3.2. Spatial variability

### 3.2.1. CDOM optical properties

CDOM absorption coefficients were calculated for the 31 sampling stations between 350 and 800 nm at 0.5 nm intervals (Fig. 4). We observed an $a_{CDOM}$ gradient across the lake (Fig. 5a) with higher $a_{CDOM}$(440) values in Kis-Balaton and the Keszthely basin (I) (where the Zala River enters into the lake) decreasing rapidly towards the northeastern basins near the outflow.

The $a_{CDOM}(440)$ coefficients were markedly different between basins ranging from 0.17 to 7.89 m$^{-1}$ with the highest value observed in Keszthely basin at the mouth of Zala River  (Fig. 5a and Fig. 6a) (Table 2). Mean $S_{CDOM}$ for the wavelength domain from 350-500 nm was 0.0206 nm$^{-1}$ ranging from 0.0161 to 0.0230 nm$^{-1}$ and with mean values per basin specified in Table 2.

$S_{CDOM}$ coefficients were relatively constant across the sampling stations (Fig. 5b, Fig. 6b), except in Kestzthely (I) basin and the western parts of Szigliget (II) basin nearest the inflow of the Zala River.  Marked variability was observed with lower $S_{CDOM}$ coefficients than elsewhere in the lake. In our study, $S_{CDOM}(350-500)$ exhibited a negative correlation with $a_{CDOM}(440)$ (Fig. 7a & b) as has been shown in previous studies (e.g., Stedmon et al. 2000, Del Castillo & Coble 2000; Yacobi et al. 2003; Rochelle-Newall et al. 2004; Zhang et al. 2007; Kowalczuk et al. 2003).

The E2:E3 ratio varied significantly between the mouth of the river (11.1) and the main outflow (62.0) as specified in Table 2.  SUVA$_{254}$ varied between 2.47 L mg$^{-1}$ m$^{-1}$ at ST25 in the Siófok basin to 4.45 L mg$^{-1}$ m$^{-1}$ at ST06 in the Keszthely basin. Figure 6d and 6e shows how E2:E3 ratio increases (R$^2$=0.470, p=0.0002) with increasing distance to the main inflow, whereas SUVA$_{454}$ decreases with distance (R$^2$=0.713, p<0.0001).

### 3.2.2. DOC

DOC data were only available for 18 out of 31 stations (Table 2). Concentrations ranged from a minimum of 8.03 at ST17 (Basin III) to 10.9 mg L$^{-1}$ at ST07 (Basin 1) with a mean value of 9.67 in the Keszthely basin (I), 8.85 for the Szigliget basin (II), 8.56 for the Szemes basin (III) and 8.66 for the Siófok basin (IV). DOC concentrations slowly decreased with increasing distance to Zala River (Fig. 5c, Fig. 6c), in similarly to the trends observed for $a_{CDOM}(440)$, but with greater variability through the system than for either $a_{CDOM}$ or $S_{CDOM}$. DOC concentrations showed a strong and positive correlation with $a_{CDOM}(440)$ coefficients over the entire dataset (Fig. 8) (R$^2$=0.945, p<0.0001).

### 3.3. CDOM photodegradation experiment

Ultraviolet irradiance during the photobleaching experiment ranged from 7.79 to 42.85 MJ m$^{-2}$ per day with a mean of 32.44 MJ m$^{-2}$ per day (Fig. 9a). The exposure of CDOM to natural solar UV radiation at these intensities, even over relatively a short exposure time, resulted in marked alterations to the optical properties of CDOM. For the dark control samples $a_{CDOM}(440)$ remained constant for the allochthonous treatment from days 0 to 7 compared with the initial value (Fig. 9c & d), indicating minimal bacterial degradation. However, the CDOM$_{allo}$ samples exposed to natural solar radiation demonstrated considerable reductions in absorption (Fig. 9c & d) at the

visible part of the spectrum (440 nm). After 7 days, $a_{CDOM}$(440) for the allochthonous samples decreased from 7.04 to 3.36 m$^{-1}$; this equates to a rate of loss of 1.49 m$^{-1}$ d$^{-1}$ and a 42 % net decrease in absorption from day 0.

In contrast, the CDOM$_{auto}$ samples showed an increase in $a_{CDOM}$(440) during the first 3 days (Fig. 9c & d), from 1.22 to 1.34 m$^{-1}$ followed by a decrease in absorption during the last 4 days, from 1.34 to 0.312 m$^{-1}$, which equates to a 77 % decrease (1.03 m$^{-1}$) from the value at day 3.

The autochthonous control samples (CDOM$_{auto-dark}$) varied in $a_{CDOM}$(440) from 1.22 to 0.596 m$^{-1}$; in the absence of photobleaching the decrease in $a_{CDOM}$(440) might be explained by residual bacterial activity (although the
samples were filtered to minimise bacterial contamination before exposure).

Photodegradation also modified the spectral slope coefficient of the samples (Fig. 9b). The values of $S_{CDOM}$ (350-500) for the CDOM$_{allo}$ samples did not show significant variation over time, varying less than 0.001 nm$^{-1}$ per day. However, for the CDOM$_{auto}$ treatment, $S_{CDOM}$ coefficients decreased conspicuously until the third day (from 0.009 to 0.005 nm$^{-1}$) and then increased with further irradiation from day 3 to day 7 until recovering to its original value
(from 0.0051 to 0.0084 nm$^{-1}$). Both the spectral slope and absorption coefficient values for CDOM$_{auto}$ were significantly lower than those for CDOM$_{allo}$.

Humic-like fluorescence as indicated by $F_n$(355) decreased gradually for the CDOM$_{allo}$ samples with increasing cumulative UV radiation and exposure time (Fig. 10a) from 41.1 to 17.5 QSU (42.6 % of the original value). Interestingly, no marked changes or clear trend was observed in $F_n$(280) (Fig. 10b), suggesting protein-like
fluorescence was less susceptible to degradation by natural solar radiation.

There was more than ten orders of magnitude difference in fluorescence intensity between CDOM$_{allo}$ and CDOM$_{auto}$ samples, presumably driven by the difference in concentration. Given the low concentrations of CDOM, after Milli-Q correction, there was no measurable fluorescence signal for the autochthonous samples.

## 4.    Discussion

**4.1. Dynamics of dissolved organic carbon**

CDOM is the coloured fraction of DOC and is often the dominant light absorbing component in lakes, particularly at blue and green wavelengths. Previous research has shown that CDOM can be responsible for up to 80 % of light absorption in Lake Balaton (Riddick et al. 2015) in spite of the fact that the lake also has high concentrations of phytoplankton and non-algal particles (NAP). The influence of Zala River in the west and the biogeochemical
gradient that it generates towards the Sió channel in the east, means that concentrations of DOC, and therefore





CDOM absorption, in Lake Balaton varies greatly through the system, particularly when inputs peak in mid-summer (from 0.169 m$^{-1}$ near the outflow to 7.89 m$^{-1}$ at the mouth of the Zala river). In comparison to published data from other large systems such as Lake Erie (O'Donnell et al. 2010; 0.19-2.0 m$^{-1}$), Lake Champlain (O'Donnell et al. 2013; 0.5-1.15 m$^{-1}$) and lakes Peipsi, Vättern and Vänern (Reinart et al. 2004; 0.33-3.82 m$^{-1}$) the

magnitude of variability in $a_{CDOM}(440)$ observed in this study was markedly greater (although these previous studies might not have captured the full range of $a_{CDOM}(440)$ variation), which is perhaps surprising given the northerly latitude of some of these lakes. However, the magnitude variability in $a_{CDOM}(440)$ observed in this study has been reported previously in other systems such as Lake Taihu (Zhang et al. 2011; 1.37 to 9.55m$^{-1}$).

Interestingly, the seasonal pattern in CDOM absorption and DOC concentration varied through the system. The

annual peak(s) in $a_{CDOM}(440)$ and DOC occurred in spring and/or autumn at the majority of stations. These peaks were broadly coincident with the annual rainfall maxima, suggesting a seasonal trend tightly coupled to the flushing of organic matter from catchment soils during high flow events. This pattern is common in many temperate and boreal lakes where DOC export from catchments is driven by the availability of flushable terrestrial carbon sources and the seasonality of precipitation and/or snowmelt. Conversely, at the station nearest to the

inflow of the River Zala the main peak in $a_{CDOM}(440)$ and DOC occurred in summer (August and July respectively) with a smaller secondary peak in DOC the autumn. The summer peak in DOC was also observed at other stations in the Keszthely and Szigliget basins, although here it was secondary to a larger autumn peak. The peak was not evident in the data from the eastern basins.

The summer peak in $a_{CDOM}(440)$ and DOC at the inflow of the River Zala was clearly related to the proximity of

the station to the mouth of the River Zala and a high input of humic-rich water from the Kis-Balaton wetland complex. In wetlands, high production of DOC can occur during the growing season due to leaching from plants and biological degradation of organic detritus (Pinney et al. 2000; Freeman et al. 2004). In our study system, this summer peak in DOC production also coincides with the annual rainfall minimum and the period of lowest flow into the lake, resulting in a concentrated input of humic-rich water and elevated CDOM absorption at the inflow

of the Zala. It is also notable that the effect of this humic-rich water from the River Zala on the biogeochemistry and light climate in Lake Balaton diminishes rapidly through the system in summer with little discernible influence beyond the western part of the Keszthely basin. This can be partly attributed to the dilution of the inflow with less humic water, but also the rapid degradation of the highly biologically and photochemically reactive DOC originating from Kis-Balaton during a period when microbial activity is high due to warm water temperatures and



UV irradiance is at its maximum. The collective residence time of the Keszthely and Szigliget basins (0.28 and
0.94 years respectively) explains why the highly humic and labile components of the DOC entering the lake from
the River Zala are largely degraded before reaching the Szemes basin with only the most recalcitrant DOC
fractions persisting beyond the westernmost basins. The resulting differences in CDOM composition are clearly
reflected in the variability in the CDOM absorption characteristics ($S_{CDOM}$, $SUVA_{254}$ and E2:E3) observed through
the system. These findings agree with other field and experimental studies which have shown that CDOM can be
rapidly degraded by photobleaching in summer (Vodacek et al. 1997; Del Vecchio & Blough 2002; Zhang, et al.
2009).

In general, $S_{CDOM}$ (350-500) variation between the stations sampled during the intensive summer campaign was
lower closer to the River Zala and higher towards the outflow (Fig 2 & Fig. 6 b). The mean (and range) in $S_{CDOM}$
(350-500) for Lake Balaton in this summer period was 0.0211 $nm^{-1}$ (0.0174 to 0.0229 $nm^{-1}$), significantly higher
than values reported in several other spatial variation studies including Lake Erie (Binding et al. 2008; $S_{CDOM}$(350-
400) 0.0161 $nm^{-1}$) and Lake Chapman (O'Donnell et al. 2013; $S_{CDOM}$(400-500) = 0.0179 $nm^{-1}$).

The values reported in these studies were more comparable to the mean slopes observed for the humic-rich waters
encountered in Kis-Balaton (0.0189 $nm^{-1}$) and the Keszthely basin (0.0199 $nm^{-1}$). However, it must be stressed
that some of this variation could in part be explained by the different wavelength ranges used in the calculation
of $S_{CDOM}$. The magnitude of spatial variability in Lake Balaton was more comparable to that reported for northern
Lake Taihu, where $S_{CDOM}$ was found to vary between 0.0127 to 0.0190, from 0.0159 to 0.0220 and from 0.0122
to 0.0174 $nm^{-1}$ for the wavelength domains 280 to 500 nm, 280 to 360 nm and 360 to 400 nm respectively (Zhang
et al. 2007) and between 0.0180 to 0.0281 $nm^{-1}$ (for $S_{CDOM}$ (280-500)) in Zhang et al. (2011).

Very few studies have investigated seasonal variation in $S_{CDOM}$ in lakes. In the present study, seasonal variation
in $S_{CDOM}$ (350-500) was greatest in the eastern basin furthest from the inflow ranging from 0.0158 $nm^{-1}$ to 0.0300
$nm^{-1}$ (Table 1, Fig. 2) with a mean annual value of 0.0205 $nm^{-1}$. The range observed in Lake Balaton was greater
than previously reported in other lake systems. Ylöstalo et al. (2014) for instance reported a mean (range) of
0.0182 $nm^{-1}$ (0.0155-0.0200 $nm^{-1}$) for 15 boreal lakes in Southern Finland within the summer months.

Indeed, there is much more variation in lakes and other optically complex river-seas systems (Kowalczuk et al.
2003; Zhang et al. 2007) than in many seas (Babin et al. 2003) in spite of the fact the latter occupy a much larger
proportion of the Earth's surface





The structural modifications in CDOM that are conveyed through variation in $S_{CDOM}$ result from the interplay between the input of allocthnonous DOC from the catchment, the production of autocthnonous DOC from the microbial digestion of phytoplankton cells and the rate at which these materials are mineralised biologically and photochemically (Zhang, et al. 2009; Yamashita & Tanoue 2004; Vantrepotte et al. 2007; Nelson et al. 1998; Yamashita et al. 2013). Newly produced autochthonous CDOM typically has a higher $S_{CDOM}$ coefficient compared to humic-rich allochthonous material. Photobleaching also results in an increase in $S_{CDOM.}$ The majority of sampling stations in Lake Balaton exhibited higher slope coefficients during the summer months, which could be attributed to an increase in new autochthonous DOC production from the growth and decay of phytoplankton during the summer bloom period and intense photobleaching of humic-rich material received from the catchment. Seasonal variability in $S_{CDOM}$ was notably lower at the inflow of the River Zala than in the easternmost basins, which again probably reflects the marked effect that intense summer photobleaching has on the structural composition of dissolved organic matter as it slow moves through the system.

Noticeably, while $a_{CDOM}(440)$ and $S_{CDOM}(350\text{-}500)$ demonstrated a strong inverse relationship over the entire dataset, the slope of this relationship varied significantly between the different basins in the lake. The slope of the relationship highest in the eastern basin and lower in the west near the inflow where DOC concentrations were the highest. The relationship between $a_{CDOM}(440)$ and $S_{CDOM}$ is known to be influenced by both the provenance and subsequent transformations (Carder et al. 1989; Helms et al. 2008). The observed trends in Lake Balaton are likely to be a result of the mixing of water rich in allocthnonous carbon from the River Zala with more dilute and autocthnonous carbon sources in the main lake and the progressive degradation of this material via photobleaching as it moves through the system. Comparable trends have been found by Zhang et al. (2007) in the Yunnan Plateau lakes.

Previous studies have also shown marked differences in the E2:E3 ratio between natural waters with differing of dissolved organic carbon. In coastal waters E2:E3 values are typically within the range 8.70±1.4 and 13.5±1.6 (nearshore – offshore in Georgia Bight; Helms et al. 2008). In inland and transitional waters, E2:E3 ratios as high as 14.6 have been reported in Lake Taihu (Zhang et al. 2011) and up to 26.9 in Chesapeake Bay (Helms et al. 2008). E2:E3 values for Lake Balaton were significantly higher and more variable than previously reported varying between 11.0 and 72.0, with the highest values near the outflow indicating the CDOM here was less humified and had a lower molecular weight. Increasing values of E2:E3 ratio have been reported by other authors indicating a decrease in colour as well as in molecular weight (Helms et al. 2008; Peuravuori & Pihlaja 1997).



Similar trends were also observed in $SUVA_{254}$. In Lake Balaton, $SUVA_{254}$ varied between 2.5 mg$^{-1}$ m$^{-1}$ in the easternmost basin and 4.5 L mg$^{-1}$ m$^{-1}$ near the mouth of the River Zala. This again indicates that water entering from the river contained high molecular weight dissolved organic carbon with a high content of aromatic

substances (Weishaar et al. 2003), whereas the compounds comprising the DOC in the central and eastern parts of the lake had a lower molecular size and aromaticity. The $SUVA_{254}$ values measured in Lake Balaton in were broadly comparable to those reported for other lake systems. For example, Song et al. (2013) reported a maximum value of $8.7 \pm 2.8$ (L mg$^{-1}$ m$^{-1}$) for 26 inland water bodies in China. $SUVA_{254}$ values were lower than those reported for marine waters where the relative contribution of autochthonous carbon sources is often greater. The

sensitive of $SUVA_{254}$ to changes in the carbon provenance is shown by Asmala et al. (2013) who obtained a range $SUVA_{254}$ values of $3.58\pm0.33$ $5.41\pm0.40$ L mg$^{-1}$ m$^{-1}$ in three Baltic Sea estuaries whereas measurements taken from stations on the sea shelf varied between $1.87\pm0.09$ and $3.47\pm0.27$ L mg$^{-1}$ m$^{-1}$.

### 4.2.  CDOM photobleaching

The spatio-seasonal variability in CDOM absorption in Lake Balaton strongly suggests that photobleaching plays

a key role in the processing and degradation of dissolved organic carbon as it flows through the system. Rapid degradation of allochthonous CDOM was observed, which was especially pronounced at the time of year with the highest solar radiation (Fig. 2) but probably also enhanced bacterial activity as a response to high water temperatures during the summer period. The processing and transformation of dissolved organic carbon by photobleaching not only influences carbon cycling, but it also is accompanied by an increase in the transparency

of the water column and changes in the optical properties that have wider implications for the underwater light climate and primary production.

The in-lake incubations conducted in Lake Balaton provide further substantiation for the critical role of photochemistry in the turnover of CDOM. We observed rapid changes in the absorption properties of CDOM in response to exposure to natural UV irradiation. In the allochthonous CDOM treatments, $a_{CDOM}(440)$ decreased

from 7.04 to 3.36 m$^{-1}$ (rate of loss of 1.49 m$^{-1}$ d$^{-1}$ and a 52 % net decrease in absorptivity from day 0) over 7 days of experiment. This rate of degradation is higher than that obtained for Lake Taihu by Zhang et al. (2013) who reported a 22 % decrease over 9 days. Bacterial degradation was not significant in the allochthonous samples as there was almost no difference in $a_{CDOM}(440)$ for the dark treatment compared to the initial value (Fig. 9d) although we cannot exclude the possibility of enhanced bacterial degradation in light exposed treatments (Kragh

et al. 2008).





The autochthonous CDOM samples exhibited an increase of $a_{CDOM}(440)$ from 1.22 to 1.33 m$^{-1}$ during the first 3 days of the experiment (Fig. 9c & 9d). This can be tentatively interpreted as short-term production of CDOM through humification of the autochthonous carbon, perhaps enhanced by photochemical processes. After three days, humification appeared to slow and CDOM degradation increased resulting in a decrease in absorption from

1.33 to 0.31 m$^{-1}$ during the final 4 days. The rate of loss was 1.23 m$^{-1}$ d$^{-1}$ equating to a 76.9% net decrease in absorptivity from day three  The rapid degradation of autochthonous observed here agrees with results reported for other systems such as for Lake Taihu where autochthonous material was also found to degrade more rapidly (Zhang et al. 2013). This rapid degradation results from the fact that terrestrially-derived allochthonous carbon is chemically more complex than autochthonous material and has a higher aromatic content, which strongly absorbs

UV and thus makes it more susceptible to photodegradation (Pullin et al. 2004; Zhang et al. 2013).

Photodegradation also modified the spectral slope (Fig. 9b) of the CDOM absorption spectra. Both the spectral slope and absorption coefficient for autochthonous CDOM were significantly lower than for allochthonous samples. During our experiment, $S_{CDOM}(350\text{-}500)$ did not follow a systematic trend in the allochthonous samples, varying less than 0.0007 m$^{-1}$ per day. However, for the autochthonous treatment, it decreased conspicuously until

the third day (from 0.0087 to 0.0051 m$^{-1}$) but then increased again from day 3 to day 7 until it almost returned to its original value (from 0.0051 to 0.0084m$^{-1}$). The increase in slope has been considered by Helms et al. (2008) to be a result of transformation from high to low molecular weight CDOM and is considered to be a response to photo-induced decomposition (Moran et al. 2000; Grzybowski 2000; Yamashita et al. 2013).  The initial decrease in slope during the early part of the experiment echoes observations by  Yamashita et al. (2013) and Fichot &

Benner (2012) who attributed this phenomena to microbial degradation of bioavailable CDOM (Nelson et al. 2004).

The fluorescence spectra also indicate a marked difference in composition between the allochthonous and autochthonous material. The decomposition of CDOM into lower molecular weight compounds under UV-B radiation (Lepane et al. 2003) results in a significant loss of both absorption and fluorescence. The negligible

fluorescence signal observed for the autochthonous CDOM samples in this study indicates it was largely comprised of low molecular weight compounds.  In contrast, the humic-like fluorescence signal measured from allochthonous samples was initially high but decreased over the experimental period from 41.07 to 17.48 QSU (57.5 % decrease). Similarly, we observed a reduction in protein-like fluorescence from 2.06 to 1.93 QSU (6.31 % decrease; Fig. 10b) over the 7 days of the experiment.  This agrees strongly with the results of previous studies

showing the fluorescence signal from humic compounds is rapidly lost through photobleaching, whereas aromatic-



like fluorescence is generally not as susceptible to degradation. Helms et al. (2013) for example reported an 84% decrease in humic-like fluorescence in response to photobleaching compared to an only 47 % decrease in aromatic-like fluorescence after 68 days of continuous irradiation in a UV solar simulator.

### 4.3. Implications for the underwater light field

The absorption of light by CDOM is a major determinant of water transparency in lakes and the availability of light for primary production. The results of this study and elsewhere clearly evidence that the dynamic nature of dissolved organic carbon in lakes results in marked spatio-seasonal variation in both the magnitude and wavelength-dependency of light absorption by chromophoric substances. This variability undoubtedly has implications not only for the quantity of light available to photosynthetic organisms but also its quality. High

concentrations of CDOM result in intense absorption of light at blue and green wavelengths but the intensity of absorption decreases exponentially with wavelength. The steep gradient in the concentration of DOC in Lake Balaton from the inflow of the Zala to the outflow of the Siófok canal will thus profoundly influence the spectral quality of light available to primary producers. This not only has implications for the productivity of the system, with the possibility of light limitation in regions with the highest absorption by CDOM (Cory et al. 2015), but

also for the photo physiology and species composition of the phytoplankton community. The intense absorption of UV light by CDOM protects phytoplankton from physiological damage and reduces the need for phytoplankton cells to manufacture UV-protective pigments. This can result in chromatic acclimation with phytoplankton in high CDOM waters investing less in UV-protective pigments (Riddick et al. 2015).

The magnitude of variability in the spectral dependency of CDOM absorption also has implications for bio-optical

models of the underwater light field that are used to underpin remote sensing algorithms for estimation of CDOM in lakes and other inland waters. Existing bio-optical models (Lee et al. 2002) commonly extrapolate absorption by CDOM in the blue to longer wavelengths using a fixed slope coefficient. Variations in the spectral-dependency of CDOM absorption can be accommodated by varying the slope coefficient based on knowledge of the optical properties of the water column. However, we demonstrate here that even within a single lake system significant

variability can occur in $S_{CDOM}$. Failure to accommodate variability in $S_{CDOM}$ in bio-optical models will lead to errors not only in the estimation of CDOM absorption but also in the contributions of other optically-active substances (e.g., chlorophyll, non-algal particles) to light absorption and scattering within the water column. In Lake Balaton, the variability observed in $S_{CDOM}$ (0.0174 - 0.0289 nm$^{-1}$) could produce errors up to 180 % and 900 % on estimates of $a_{CDOM}$ at wavelengths in the blue (350 nm) and red (650 nm) respectively. This suggest that





new approaches are need to incorporate knowledge on the variability in $S_{CDOM}$ into adaptive bio-optical modelling

frameworks for optically-complex waters to improve our ability to model the underwater light field and increase

the performance of physics-based remote sensing algorithms for CDOM retrieval.

## 5.    Conclusions

This study revealed the high spatial and seasonal variability in the quantity and quality of CDOM that can exist

within a large, temperate shallow lake.  The variation was strongly driven by the allochthonous input of dissolved

carbon from the Zala River and its rapid transformation as it moves through the system. The variability in the

quantity and quality of CDOM was strongly reflected in a number of readily measured optical parameters

including $S_{CDOM}$, E2:E3 ratio and SUVA$_{254}$, which collectively pointed towards a marked decrease in the

molecular weight of dissolved carbon compounds, a reduction on its aromatic content and a decrease in the degree

of humification as water moved through the system from the main inflow to outflow.

Photobleaching was found to be a major factor controlling the in-lake transformation and degradation of CDOM,

and a key process influencing the spatial structure CDOM throughout the system.  The photobleaching rate

coefficient for allochthonous CDOM was found to be higher than for autochthonous CDOM due to the greater

photoreactivity of terrestrially-derived compounds.  CDOM in Lake Balaton is mainly terrestrial in origin and is

thus rapidly degraded by exposure UV irradiation.  The implied importance of photobleaching to carbon dynamics

is consistent with previous studies conducted in other inland water bodies (Zhang et al. 2013) as well as other

studies carried out in shelf seas (Babin et al. 2003) and the open ocean (Helms et al. 2013).

More widely, these results provide novel insight on the potential contribution of wetlands to DOM and CDOM in

lakes, not only in terms of the concentration of CDOM but also its seasonality. The seasonal trend in CDOM

observed close to the main inflow was significantly different from that observed elsewhere in the system.

Notwithstanding the fact that most of the CDOM in Lake Balaton would seem to be terrestrial in origin, we did

observe an increase in $a_{CDOM}$ (440) in autumn following the breakdown of phytoplankton blooms on the lake.

The observed spatial and temporal variability in the optical properties of CDOM in this study has important

implications for biogeochemical cycling in Lake Balaton but also for bio-optical models of the underwater light

climate in lakes and their application in the parametrization of algorithms for optical remote sensing of CDOM

and other optically-active constituents.



## 6. Data availability

The data analysed in this study will be freely accessible through the University of Stirling DataSTORRE Open Access Repository.

*Author contributions*

M.E. Aulló-Maestro designed and conducted the experiments with input from P. Hunter and E. Spyrakos. Hajnalka Horváth and Mátyás Présing carried out data collection for seasonal measurements at Balaton Limnological Institute. Spatial measurements were taken by GloboLakes and INFORM teams. Jesús M. Torres Palenzuela contributed to fluorescence measurements at University of Vigo and Tom Preston contributed to mass-

spectrometry analysis at the Stable Isotope Biochemistry Laboratory at the Scottish Universities Environmental Research Centre (SUERC). M.E. Aulló-Maestro processed the data and prepared the figures with input from Pierre Mercatoris. M.E. Aulló-Maestro prepared de manuscript with the assistance of all co-authors.

*Acknowledgements*. The authors gratefully acknowledge funding from the UK NERC funded GloboLakes project (REF NE/J024279/1), the EU FP7 INFORM project (Grant Agreement Number 606865) and the Hungarian

Academy of Science TÁMOP-4.2.2 A-11/1/KONV-2012-0038 project. Fluorescence measurements at University of Vigo were funded by a Santander doctoral travel award. Field and lab assistance on Lake Balaton was provided by Laura Ulsig, Viktória Horváth, Anett Kelemen, Eszter Zsigmond, Piroska Rádóczy, Anna Kánicz and Ádám Szigethy. Lab assistance at University of Vigo was provided by Marta Iglesias Trigo. María E. Aullo-Maestro was supported by GloboLakes project and a University of Stirling impact studentship. The authors would like to

particularly emphasize their acknowledgement to the late Matyas Presing, a valued mentor and friend who greatly contributed to this research.



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



**Tables**

**Table 1. Values of CDOM absorption coefficient at 440nm, CDOM slope coefficient between 350 and 500nm, DOC concentration, E2:E3 ratio, SUVA254 and mean distance of the basin to River Zala. Values obtained for CDOM seasonal variation.**

| Station | Basin | $a_{CDOM}(440)$ $(m^{-1})$ Max-Min (Mean-SD) | $S_{CDOM}(350-500)$ $(nm^{-1})$ Max-Min (Mean-SD) | [DOC] mg·L$^{-1}$ Max-Min (Mean-SD) | Distance to River Zala (Km) |
|---|---|---|---|---|---|
| 01 | Keszthely | 9.01-3.69 **(6.52-1.54)** | 0.0181-0.0161 **(0.0173-0.0006)** | 19.70-10.02 **(16.08-2.88)** | 0.48 |
| 03 | | 1.43-0.64 **(0.94-0.28)** | 0.0221-0.0187 **(0.0201-0.0011)** | 12.67-8.97 **(10.15-1.04)** | 3.59 |
| 12 | Szigliget | 0.63-0.42 **(0.53-0.09)** | 0.0230-0.0185 **(0.0211-0.0016)** | 10.85-8.88 **(9.51-0.66)** | 14.95 |
| 20 | Szemes | 0.61-0.15 **(0.37-0.15)** | 0.0275-0.0175 **(0.0213-0.0035)** | 9.00-8.23 **(8.76-0.23)** | 39.73 |
| 25 | Siófok | 0.26-0.09 **(0.20-0.06)** | 0.0279-0.0185 **(0.0215-0.0033)** | 8.54-7.63 **(8.13-0.25)** | 62.86 |
| 30 | | 0.33-0.06 | 0.0300-0.0158 | 8.36-7.82 | 69.16 |





**Table 2.** Values of CDOM absorption coefficient at 440nm, CDOM slope coefficient between 350 and 500nm, DOC concentration, E2:E3 ratio, SUVA254 and mean distance of the basin to River Zala. Values obtained for CDOM spatial variation.

| Basin | $a_{CDOM}(440)$ (m$^{-1}$) Max-Min (Mean) | $S_{CDOM}(350-500)$ (nm$^{-1}$) Max-Min (Mean) | [DOC] mg·L$^{-1}$ Max-Min (Mean) | E2:E3 ratio Max-Min (Mean) | SUVA$_{254}$ (L·mg$^{-1}$·m$^{-1}$) Max-Min (Mean) | Mean distance to River Zala (Km) |
|---|---|---|---|---|---|---|
| | **(0.19-0.10)** | **(0.0219-0.0049)** | | **(8.01-0.18)** | | |
| Kis-Balaton | 4.66-2.45 **(3.58)** | 0.0191-0.0186 **(0.0189)** | --- | | --- | |
| Kestzthely | 7.89-0.57 **(1.49)** | 0.0212-0.0174 **(0.0199)** | 8.85-10.9 **(9.66)** | 11.1-34.1 **(15.9)** | 3.59-4.45 **(4.04)** | 3.84 |
| Sziglget | 4.31-0.33 **(1.00)** | 0.0214-0.0190 **(0.0209)** | 8.50-9.63 **(8.85)** | 14.4-28.8 **(18.9)** | 3.24-4.35 **(3.70)** | 17.4 |
| Szemes | 0.34-0.26 **(0.294)** | 0.0221-0.0211 **(0.0215)** | 8.03-9.07 **(8.56)** | 18.4-44.3 **(27.2)** | 2.62-3.82 **(3.12)** | 40.0 |
| Siófok | 0.21-0.17 **(0.193)** | 0.0229-0.0203 **(0.0215)** | 8.14-8.99 **(8.66)** | 24.1-62.0 **(42.2)** | 2.47-3.13 **(2.69)** | 60.5 |





**Figures**

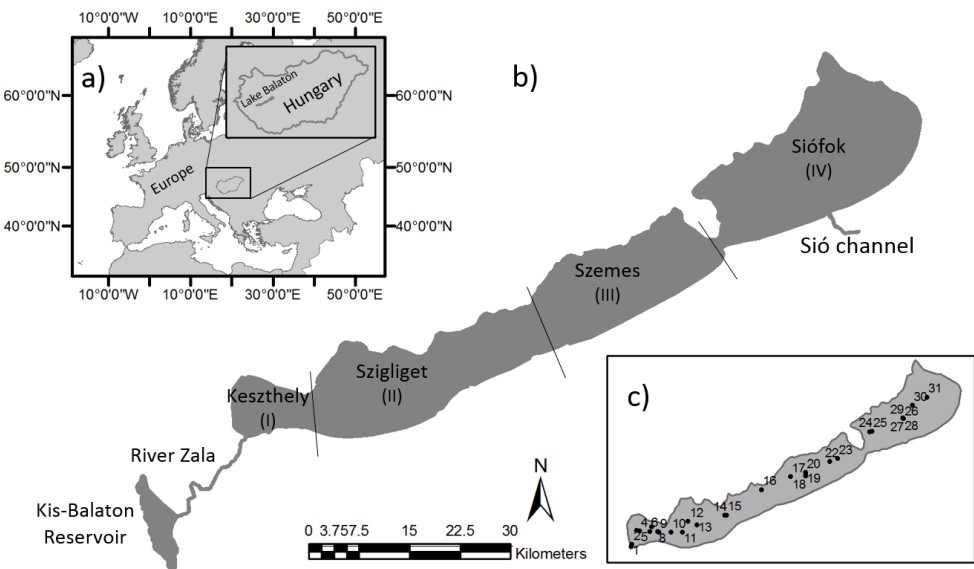

Figure 1. a) Location of Lake Balaton within Europe. b) Map of basins, Kis-Balaton Reservoir, River Zala and Sió Channel. c) Location of 31 sampling stations in Lake Balaton

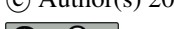



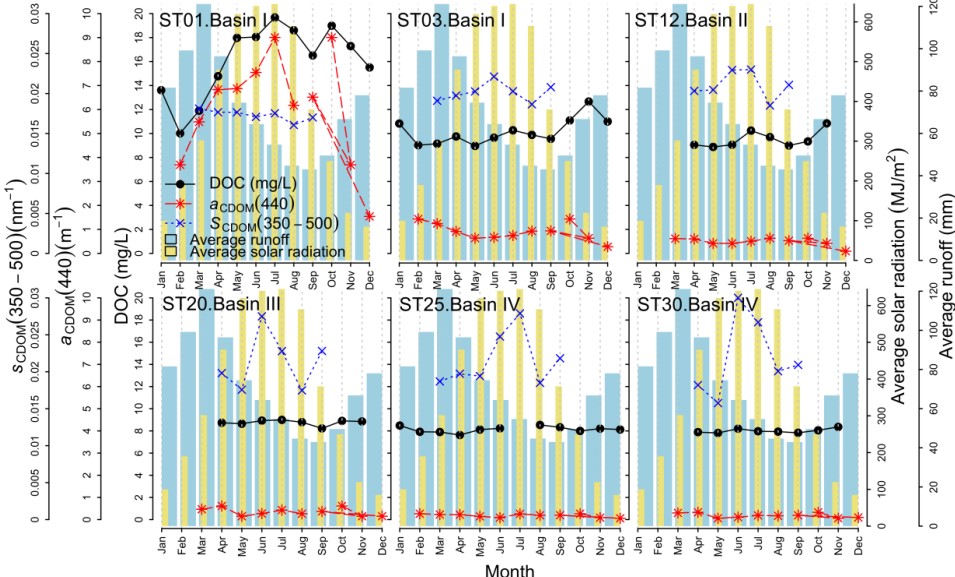

Fig. 2 Seasonal aCDOM(440), SCDOM(350-500) and DOC concentration variation in Lake Balaton between January and December 2014 and seasonal variability of runoff in Balaton region (Hungary), monthly means from 1921 to 2007 (modified from Anda & Varga, 2010).




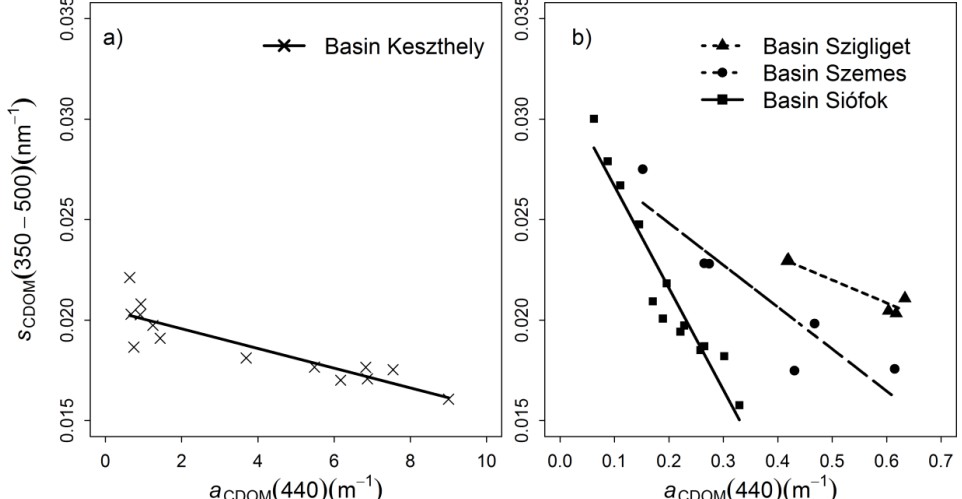

**Fig. 3. Plot of $S_{CDOM}$ (350-500) as a function of $a_{CDOM}$ (440) using the seasonal sampling data for (a) basin Kesthely, $S_{CDOM}$(350-500) = -0.0005·$a_{CDOM}$(440) + 0.0205, $R^2$=0.7833, p<0.0001 and (b) basin Szigliguet, $S_{CDOM}$(350-500) = -0.0114·$a_{CDOM}$(440) + 0.0277, $R^2$=0.9122, p=0.011; basin Szemes, $S_{CDOM}$(350-500) = -0.0209·$a_{CDOM}$(440) + 0.0209, $R^2$=0.7932, p=0.0108 and basin Siófok, $S_{CDOM}$(350-500) = -0.0507·$a_{CDOM}$(440) + 0.0317, $R^2$=0.9154, p<0.00001.**




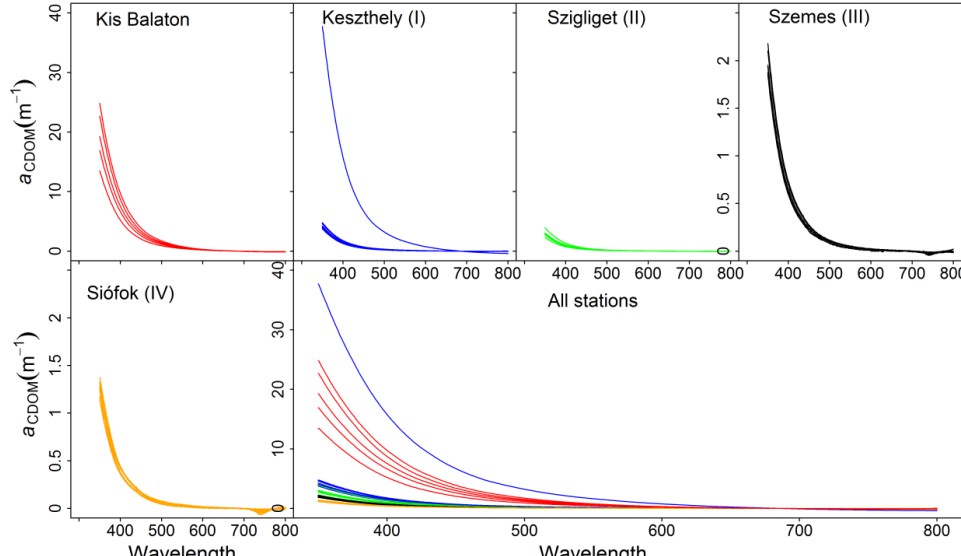

**Fig. 4 CDOM absorption spectra for all stations (per basin) and Kis-Balaton. Note the different y-axis scale for basins Szemes and Siófok.**





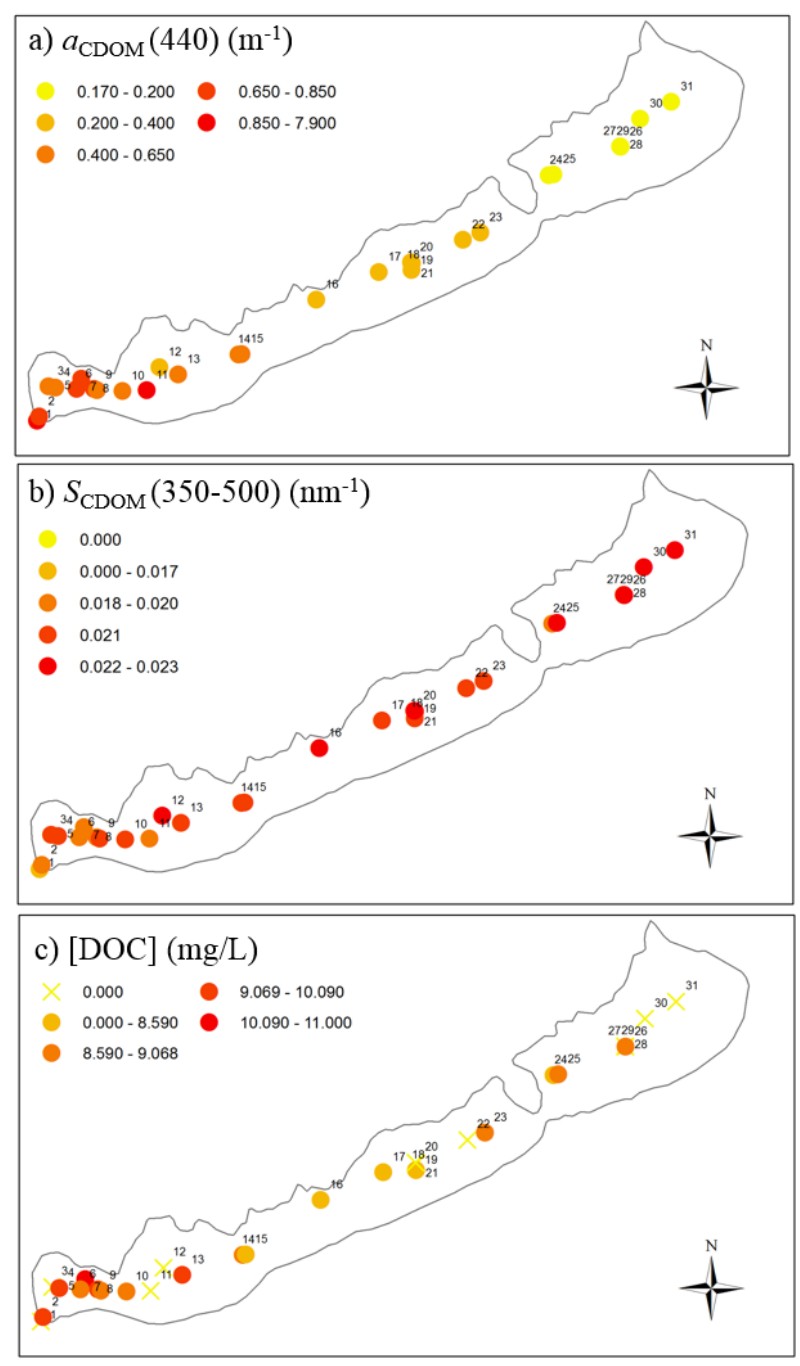

**Figure 5. a) Spatial $a_{CDOM}$(440) variation in Lake Balaton per station. b) Spatial $S_{CDOM}$(350-500) variation in Lake Balaton per station. c) Spatial DOC concentration in Lake Balaton per station**



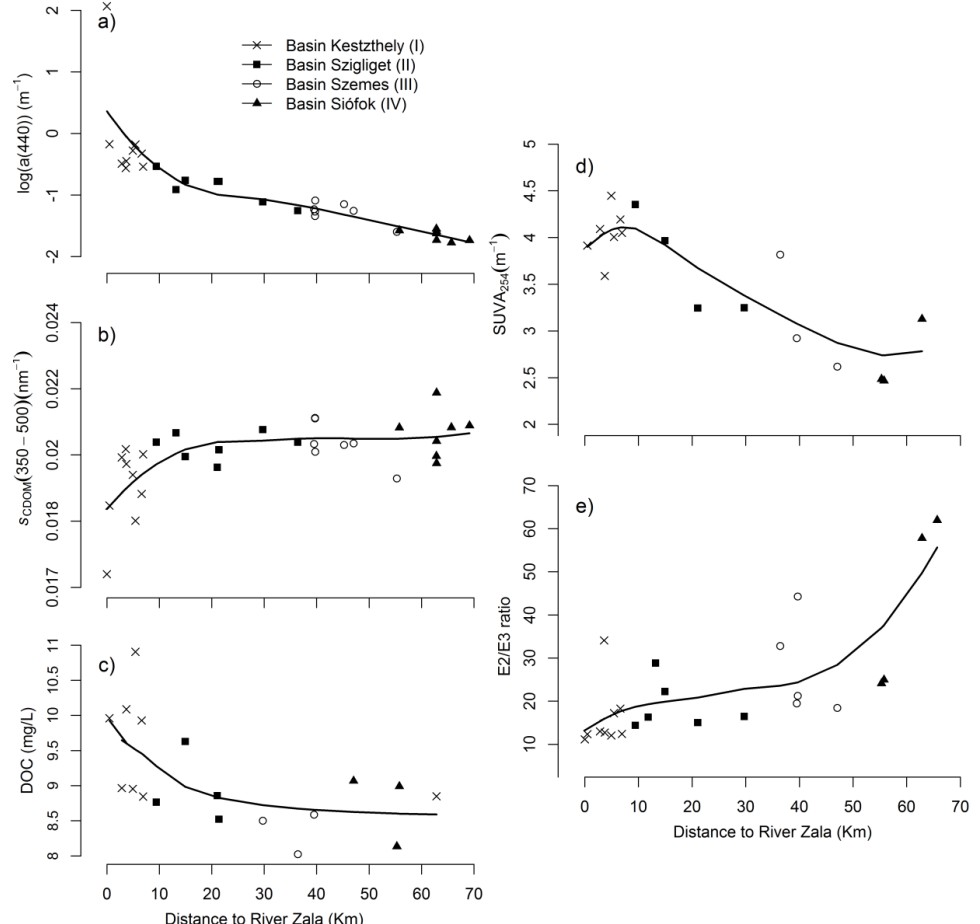

**Figure 6. Scatterplots against distance to the main inflow [Km] with loess curve fitted to data. (a) Variation of CDOM absorption coefficient at 440 nm ($a_{CDOM}$ (440)) [m$^{-1}$], (b) CDOM slope coefficient between 350 and 500 mm ($S_{CDOM}$ (350-500)) [nm$^{-1}$] variation, (c) DOC concentration [mg/L] variation, (d) specific UV absorptivity at 254nm (SUVA$_{254}$) [m$^{-1}$] and e) E2/E3 ratio as a function of distance from the Zala River during the summer 2014 campaign.**



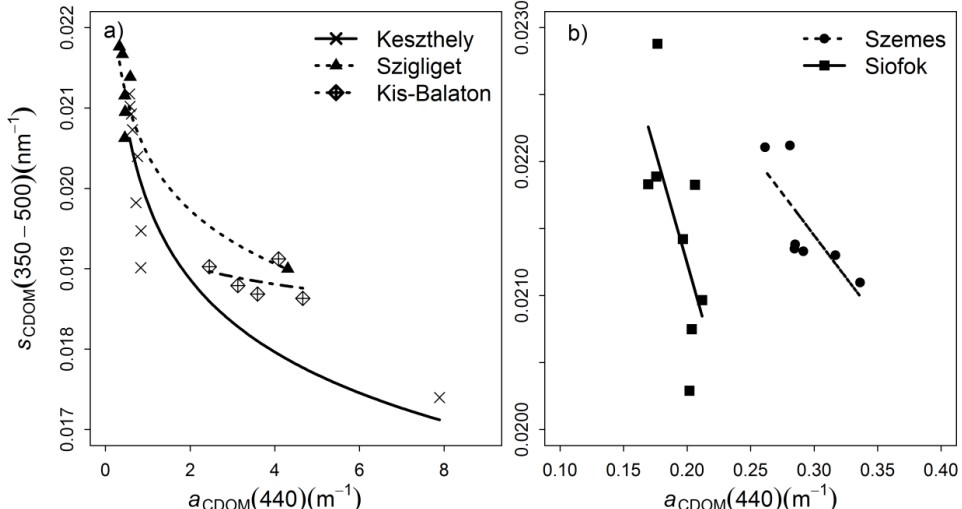

**Figure 7. Plot of** $S_{CDOM}(350\text{-}500)$ **as a function of** $a_{CDOM}(440)$ **spatial variation. a) Kis Balaton,** $S_{CDOM}(350\text{-}500) = 0.019266 \cdot a_{CDOM}(440)^{-0.017362}$; **basin Keszthely,** $S_{CDOM}(350\text{-}500) = 0.019817 \cdot a_{CDOM}(440)^{-0.070820}$ **and basin Szigliget,** $S_{CDOM}(350\text{-}500) = 0.020418 \cdot a_{CDOM}(440)^{-0.070820}$. **b) Basins Szemes,** $S_{CDOM}(350\text{-}500) = -0.01252 \cdot a_{CDOM}(440) + 0.02521$ **and Siofok,** $S_{CDOM}(350\text{-}500) = -0.03330 \cdot a_{CDOM}(440) + 0.027900$.





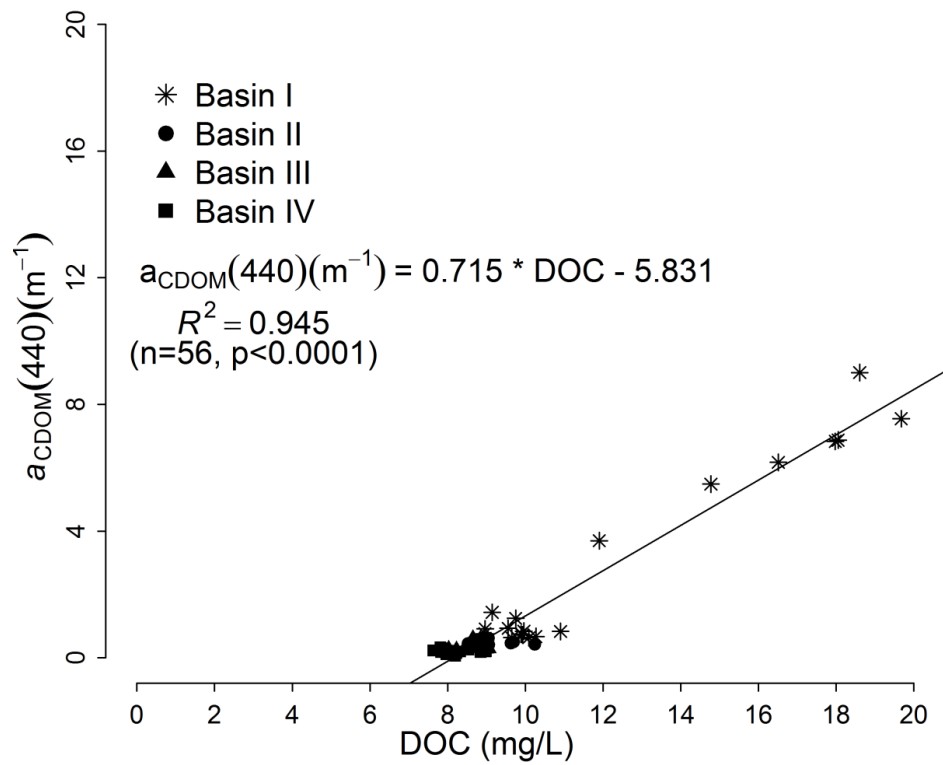

**Figure 8. Scatterplot of $a_{CDOM}(440)$ plotted as a function of DOC concentration (mg/L). Line is a regression curve by**

**least squares fit.**



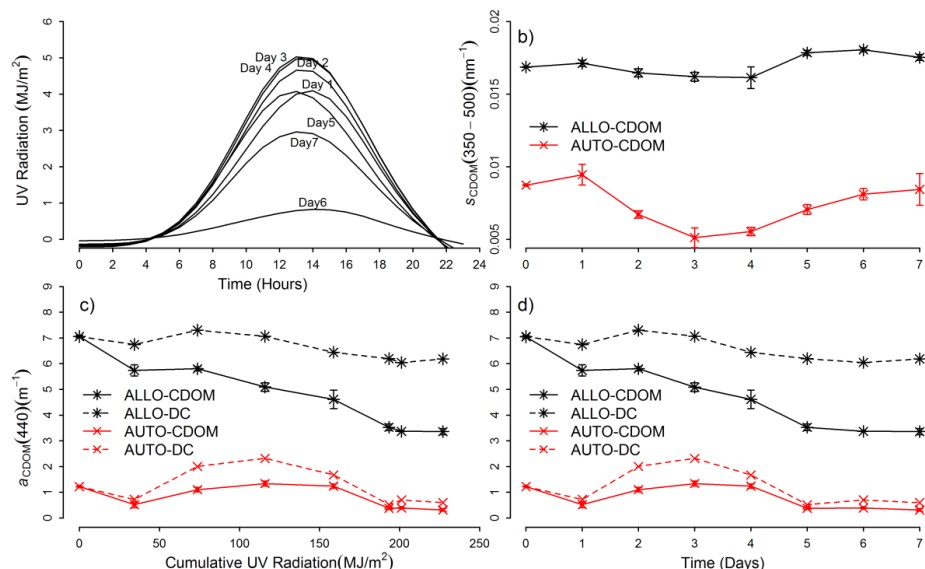

**Figure 9. a)** Ultraviolet irradiance during the photobleaching experiment. **b)** Variation of $S_{CDOM}$ **(350-500)** per day. **c)** Variation of $a_{CDOM}$ **(400)** accumulated UV radiation. **d)** Variation of $a_{CDOM}$ **(400)** per day. **Note that the error bars represent ± standard deviation and exist for every data ponint corresponding with exposed samples in sub-figures b), c) and d), dark samples not included.**

845




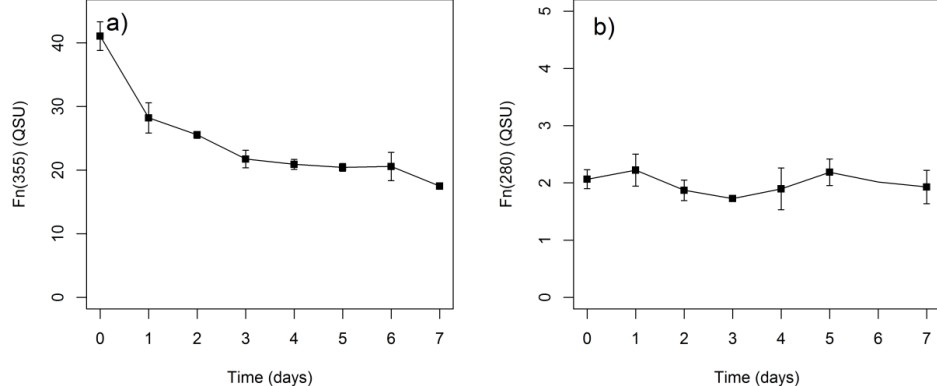

**Figure 10. Changes of Humic-like fluorescence (Fn(355)) and protein-like fluorescence (Fn(280)) for allochthonous CDOM samples with time during photobleaching experiment. Bars = ±Standard Deviation.**