# Peer review of "Spatio-seasonal variability of chromophoric dissolved organic matter absorption and responses to photobleaching in a large shallow temperate lake"

_Biogeosciences, 2016_

## Referee Comment (RC1) · Anonymous Referee #1 · 5 Sep 2016

Reviewer comments on Biogeosciences Discuss., doi:10.5194/bg-2016-329, 2016

General comments. This paper documents variability in DOM along a longitudinal transect in Lake Balaton during 2014. The authors report that DOM concentrations generally ranged from 8 to 16 mgC/L in surface waters (which is relatively high), with values above 10 mgC/L mostly in the eastern end of the lake which receives inflow from a large wetland. They also report four optical properties of the DOM (absorbance, spectral slope coefficient, SUVA and E2/E3) as potential indices of DOM source or internal processing (especially photodegradation). Unfortunately, the choice of wavelengths for some of the optical measurements was not optimal, thus limiting comparison with other systems and studies. Although the paper reports apparently new information about Lake Balaton, there are several matters in the text, tables and figures that will require re-thinking and major revision.

Specific comments.

1. Abstract. Line 35. The data on Fig 9c and 9d are not convincing evidence that UV irradiation caused any change in the absorbance of "autochthonous" DOM. Omit sentence
2. Abstract. Omit last two sentences. Nothing new here.
3. Lines 70-73. Unclear sentence. What nutrients are we talking about? Re-think and re-write.
4. Line 93. The wavelength 440nm is not commonly used for CDOM. 440nm is routinely used for Chl-a because it is the absorbance maximum for that pigment. The preferred wavelength for CDOM is in the UV, usually less than 370nm. Choice of 440 (blue) rather than UV needs to be justified.
5. Line 95. The spectral slope coefficient recommended by Helms et al (2008) and by Fichot and Benner (2012) is calculated over the UV range 275-295 nm. But in Balaton, the authors used a very different range (350-500 nm). Choice of this range needs to be justified given that the authors cite Helms et al and Fichot and Benner.
6. Line 113-120. Not true. There is a relatively rich literature on DOM in temperate lakes. Literature search needed.
7. Lines 124-127. For objective #1, add "over the course of one year."  For objective #3, since there were no direct measurements of the underwater light field (e.g.Kd PAR, or spectral Kd), this is not a bona fide objective.
8. Line 174-175. One year of data is not sufficient to document a representative seasonal cycle. Caveat needed.
9. Line 180. A 0.7um filter will not remove small phytoplankton, bacteria or large organic colloids. Explain why 0.2um filters were not used, and what the consequences of including small particulates might be.
10. Lines 211-212. Helms et al (2008) recommended S275-295 as an indicator of in situ DOM photoprocessing for future DOM studies (not E2:E3). That needs to be acknowledged.
11. Section 2.5. CDOM photodegredation. Except for Comment #13 (below), this is a very interesting and well planned experiment. Kudos.
12. Lines 226-234. In nature, autochthonous DOM is more likely the exudates from live phytoplankton rather than dead cell remains (which would be colonized by microbes and sink out). Since exudates would be in the supernatant not the pellet, the method of grinding and digesting the pellet needs to be justified.
13. Section 3.1. Seasonal variability. Caveat needed to acknowledge that seasonal changes were characterized for only 1 year, and may not represent the "average" seasonal cycle across multiple years with different weather patterns

14. Line 263. Rephrase sentence. With the exception of STO.Basin 1, the data on Fig 2 do not indicate high CDOM variability across most of the lake. In 5 of the 6 basins, DOC and a(440) were relatively stable.
15. Line 286. The evidence for two peaks in DOC on Fig 2 is weak at best. Rephrase.
16. Line 296. Add "during July when interbasin differences were likely to be highest."
17. Lines 303 to 308. This paragraph needs to be re-thought. The range of Scdom across stations is actually quite wide in the scheme of things for natural waters. Compare to Helms et al (2008) and other studies where Scdom gradients have been reported.
18. Line 318. Re-think. Concluding that DOC varied more than these two optical properties seems a consequence of scaling rather than a property of the variable. Comparison of CVs or Z-scores is needed here
19. Section 3.3. Photodegredation experiment. This section needs to be re-written. The results shown on Figure 9 indicate that UV irradiation had a significant effect only on the absorbance of allochthonous cdom. This result is consistent with the data on Fig 10a. That's all that one can say with confidence about the photodegredation experiment. There is no discernible effect on Scdom or on "autochthonous" cdom in these data.
20. Section 4. Discussion. This section has a some overstatement, speculation and misinterpretation that needs to be removed .
    a. L 356-363. Actually, DOM across most of the lake is relatively constant (Fig 2&5c), as observed in other large lakes. High DOM in Balaton was restricted to stations near the inflow from a high DOM river. In studies of other large lakes, such stations were likely not sampled because they skew the data and contribute little to the total lake mass of DOM.
    b. L364-374. It is an exaggeration to say that aCDOM and DOC varied seasonally throughout the system. This was only true for Basin I. In the other 4 basins, aCDOM and DOC were relatively constant throughout the year (FIG 2)
    c. L413-424. Photomineralization would not affect Scdom, it would only affect DOM concentration. But photobleaching can have a large effect on Scdom (e.g.Helms et al, 2008), and the data for Lake Balaton shown on figs 9 and 10 suggest that the humics are the fraction of DOM that is being bleached.
    d. L457. Again, microbes mineralize (respire) DOM or turn it into biomass rather than bleaching it
    e. L471-480. This is an over-interpretation of the results shown on Fig 9. It is unsupported speculation. Omit paragraph
    f. L481-491. This is also over-interpretation and speculation. In fig 9c, there was no difference between the irradiated "autochthonous" samples and the dark controls. In 9b, there were no controls. Omit paragraph
    g. L492-500. The low fluorescence signal from "autochthonous" DOM is likely due to its low concentration. Re-write
    h. L 505-519. Absent measurements of the underwater light field in the lake, it is speculation to propose light limitation by DOM in such a shallow, well-mixed and eutrophic lake. Ten percent of incident solar irradiance is generally sufficient to support strong phytoplankton growth, and the authors would need to show that attenuation by DOM was sufficient to reduce downwelling light penetration below that level in the epilimnion. Re-write paragraph
21. Table 1. Change notation to (mean±SD). Specify date range in heading
22. Table 2. Are these data for July 2013 or July 2014 (or both)? If both, how was interannual variation accounted for. Specify dates in heading

23. Fig 2. Check the trace for aCDOM(440). It behaves weirdly during autumn in some of the plots. Needs to be fixed
24. Fig 5. Add "measurements made in July…" to legend. In Fig 5c the 5 DOC categories exaggerate actual spatial variability. The three highest categories are separated by less than 1 mgC/L, while the low category spans more than 8 mg.C/L. The category indicated by a yellow X is highly questionable (zero DOC? I don't think so). Re-think this.
25. Fig 9. Panel 9d is superfluous. Omit

---

## Referee Comment (RC2) · Anonymous Referee #2 · 11 Sep 2016

General comments: The study has investigated the spatial and seasonal variation of chromophoric DOM in a lake at several stations leading away from an inflow from a River supplying allocthonous DOM. Various methods were used to capture change in DOM over time and spatially including analysis of the absorption coefficient spectral slope, E2:E3 ratio, SUVA and Spectral Fluorescence signals. The study shows that the absorbance coefficient and DOC values were highest and most variable near the inflow to the lake, whereas other basins further located from the inflow had more stable values with more seasonal variability in the spectral slope of CDOM.

While the study has performed what seems like a well-planned study there are some very strong statements that over reach the scope of the study. I suggest major revisions especially of the discussion.

Specific comments:

Line 84: What platforms are these? Please explain.

Line 86: Please remove "the" in the part of the sentence which reads "However, CDOM is the arguably most challenging. . ."

Line 118: There are quite a few studies on temperate lakes, including seasonal work, see Müller et al 2014 "Hourly, daily and seasonal variability in the absorption spectra of chromophoric dissolved. . .."

Line 174: Mention the time of the year samples occurred to represent seasonal variability.

Line 187: Instead of referring to the summer campaign as "intensive summer campaign" change to "spatial variability" in the whole manuscript.

Lines 190 and 193: Explain why two different instruments were used for the different campaigns.

Line 194-195: It is not clear when the reference sodium azide was used. Please make this clear.

Line220: What was the temperature in the lake during this 7 day incubation?

Line 211: Instead of writing "this wavelength" specify which wave length "this" refers to.

Line 233: Was CDOM measured to know the start value? Give further explanation.

Line 263: A seasonal variability in aCDOM, which was used to determine change in CDOM quantity, is not clear from figure 2. Either present statistical data backing this up or rephrase the statement.

Line 263-269: The seasons are not shown in table 1, so the information that is referred to cannot be seen in the table. Add this information to the table.

Line 270: What is the relevance of comparing August values of sCDOM with June? Do the authors mean that these are the lowest and highest values? Please make this statement more clear.

Line 292: When stating something is significantly lower the statistical data must be presented. Please add this data.

Line 295: This sentence regarding figure 4 does not present data and should be part of the methods section.

Line301: Are the values of min and max mentioned in the text also in the table? Please review.

Line304: can this statement that there was a marked variability be made with a change of what seems to be of 0,002 on a nm scale?

Line 309: Where is the statistical data backing up the statement that it "varied significantly"? Please add this data.

Line 310-311: Refer to table 2 for the SUVA data.

Line 314: this statement about DOC data availability should be in methods since this cannot be seen in table 2 it is misleading to refer to it.

Line 319: Is the correlation significant for all basins? It seems like Basin I has a strong correlation, how would it look like if they were analyzed separately?

Line 324: where is the data for these "marked alterations"? Does this refer to aCDOM ? Rewrite and connect the sentences better.

Line 325: What was the temperature during this incubation? Can you really be sure that there was no bacterial degradation, 7 days is a long time for bacteria to degrade

DOM although you filtered through 0.2$\mu$m there are always some bacteria that are small enough to get through and grow to higher abundance over time, perhaps a portion of the DOM that cannot be measured with aCDOM was taken up like what is shown in figure 10?

Line 327: where the reductions statistically significant?

Line 330: Why was there an increase in the Dark controls? Please discuss this.

Line 337: When stating no "significant variation" this implies statistical significance and thus data has to be presented. Present statistical data.

Line 341: same requirement as the previous comment. Show statistical data.

Discussion section:

Line 362: Why is it surprising that the range has not been captured in the northern latitudes? Please explain.

Lines 364-365: This statement is contradictory to your results. From figure 2 it rather seems like the aCDOM and DOC values were quite stable in most basins with variation only in Basin I at station 1 probably due to the inflow of the Zala River. Please re-write this part.

Lines 365-367: This correlation was not shown in the results and also does not seem consistent in all basins. Where is the data for this statement?

Lines 369-373: Re-write this statement since it bases its argument on the previous statement that there could be coupling between aCDOM and DOC due to rainfall events, which was not observed in this study.

Line 372: Isn't the Keszthely basin the same basin that is closest to the inflow of the Zala River and thus repeating what was stated in the previous sentence?

Line 385-386: Please add the reference for the water residence time.

Line 395: Please add the statistical data to back-up the statement made that it was "significantly higher".

Lines 395-397: This information belongs in results since it is not a discussion.

Line 398: Which studies are referred to in the statement "these studies"?

Line 418: Some references needed here about photobleaching and sCDOM, this sentence seems lost here.

Line 428-429: Please complete the sentence "influenced by both the provenance and subsequent transformations. . .." of what?

Lines 454-457: I'm not convinced that this was due to photobleaching, this section refers to figure 2, however this figure does not back-up this claim how do you rule out a dilution effect? Re-phrase.

Line 461: Please add a reference to this paragraph.

Line 465: This data needs to be compared with the control and statistical confirmation presented in the results section.

Line 481: Here if referring to allochtonous it should be less susceptible instead of more. Please change.

Line 481: There is no visible change in SCDOM in the ALLO-CDOM. Please rephrase this statement

Line 481-482: Where is the statistical data to back-up the claim of statistical significance? Is this a comparison between allocthonous with autocthonous or with start values? Please add the data to the results section and re-phrase this discussion based on this.

Line 492: Where is the data for fluorescence spectra of autocthonous material? Figure 10a an10b only present allocthonous.

Line 495: where is this data?

Line 500: could this loss not be due to bacterial degradation?

Line 505-506: Please add a reference to this sentence.

Line 506: what is meant by "elsewhere"?

Line 505-509: This is a very strong statement that cannot be proven with the data from this study. Please re-write.

Line 512: Also this statement is too bold since this was not within the scope of this study.

Line 522-524: Please add a reference to this statement.

Line 547: Isn't the contribution of wetlands well known? Remove "novel".

Technical comments:

Line 70-71: Please review this sentence, it seems like information is being repeated and there is a misuse of the word "whilst".

Line 71: In the same sentence as the above comment "...this fulfilling important role..." should probably be "thus fulfilling an important role".

Line 75: can CDOM have a behavior? Perhaps property could be used instead.

Line 87: should be changed to "for reliable estimation of remotely..." Please change.

Line 89: should be changed to "studies have explored the application..."

Line 97: change to "size of DOM molecules..."

Line 98: I think the authors mean larger/greater molecules, not higher.

Lines 131-133: Please add references to this information about the study area.

Line 136: should be changed to "...at that time of the year..."

Lines 162-164: Please add a reference to this statement.

Line 165: what is meant by "...less noticeable..."? Less than what?

Line 219: I suggest moving "fifty-six" to Line 222 so it reads "Fifty-six samples were taken in total of which 21 were composed of..."

Line 228: Please add a reference to the dominance of the phytoplankton in this particular lake.

Line 311: if reference to figure 6d and 6e is made then SUVA should be mentioned first and then E2/E3 ratio to be consistent. Then you can say that it refers to those figures respectively.

Line 315: mean value in table 2 for Keszthely basin is 9.66 not 9.67 as it says in the text. Which is correct? Please review.

Line317: Do you mean with increasing distance from Zala River?

Line 317: remove "in" before the word similarly.

Line346: Change to "there were more than ten orders..."

Reference list: I have not checked the reference list. Line434: How does this statement connect with the data in this study: "previous studies have also found marked differences in the E2:E3 between natural waters..." Present the data from the study and then connect with what other studies have found. Line446: Remove "in" after Lake Balaton. Line450: change "sensitive" to sensitivity. Line456: Add: and, between the two ranges. Line530: change to "new approaches are needed..."

Tables 1 and 2: Is there a reason why values are stated as Max-Min instead of Min-Max? Consider changing to better fit with standard way of reporting such values. Figure2: the lines connecting data points for aCDOM seem to connect in a strange way or to be disconnected. Please review and fix. Figure 9: add to legend what "DC" refers to, dark control?

---

## Author Comment (AC1) · 5 Nov 2016

Response to Reviewers – Aulló-Maestro et al. Biogeosciences Discuss., doi:10.5194/bg-2016-329, 2016 Anonymous Referee #1 General comments. This paper documents variability in DOM along a longitudinal transect in Lake Balaton during 2014. The authors report that DOM concentrations generally ranged from 8 to 16 mgC/L in surface waters (which is relatively high), with values above 10 mgC/L mostly in the eastern end of the lake which receives inflow from a large wetland. They also report four optical properties of the DOM (absorbance, spectral slope coefficient, SUVA

and E2/E3) as potential indices of DOM source or internal processing (especially pho-todegradation). Unfortunately, the choice of wavelengths for some of the optical mea-surements was not optimal, thus limiting comparison with other systems and studies.

We agree with the reviewer that this can limit the comparison with other systems and studies, however, and as justified after in the text, this paper is strongly motivated by the understanding of the changes on the inherent optical properties of CDOM particu-larly with reference to the implications for remote sensing. In this framework, remote sensing studies often use 412 nm or 440 nm (Carder et al. 1989; Nelson et al. 1998; Schwarz et al. 2002) to describe CDOM absorption because information in the UV cannot be obtained from space. For this reason, and to ensure consistency with previ-ous publications, our results are more strongly based on absorption in the blue rather than the UV.

Although the paper reports apparently new information about Lake Balaton, there are several matters in the text, tables and figures that will require re-thinking and major revision.

We wish to thank the reviewer for this very thorough and constructive review. We agree with most of the suggestions and have therefore modified the text according to them. We think they have greatly enhanced the quality of the paper.

Specific comments. 1. Abstract. Line 35. The data on Fig 9c and 9d are not convincing evidence that UV irradiation caused any change in the absorbance of "autochthonous" DOM. Omit sentence R1. The sentence has been omitted as suggested 2. Abstract. Omit last two sentences. Nothing new here. R2. The sentence has been omitted as suggested 3. Lines 70-73. Unclear sentence. What nutrients are we talking about? Re-think and re-write. R3. We agree it sounded like an incomplete sentence and have therefore rephrased it to now read as: "CDOM can be remineralised by bacteria acting as a source of inorganic nutrients (Buchan et al. 2014), which is important for phytoplankton nutrition" 4. Line 93. The wavelength 440nm is not commonly used

for CDOM. 440nm is routinely used for Chl-a because it is the absorbance maximum for that pigment. The preferred wavelength for CDOM is in the UV, usually less than 370nm. Choice of 440 (blue) rather than UV needs to be justified. R4. As justified later in the text (lines 196–198) and explained before, the fact of this study being strongly motivated by the understanding of the changes on the inherent optical properties of CDOM particularly with reference to the implications for remote sensing and to ensure consistency with previous publications, our results are based on CDOM absorption measurements at 440 nm. 5. Line 95. The spectral slope coefficient recommended by Helms et al (2008) and by Fichot and Benner (2012) is calculated over the UV range 275-295 nm. But in Balaton, the authors used a very different range (350-500 nm). Choice of this range needs to be justified given that the authors cite Helms et al and Fichot and Benner. R5. We agree our choice needs to be justified and as modified in the text (line 201-203): "This range of calculation was consistent with Babin et al. 2003 and Matsuoka et al. 2012 amongst others and is more relevant to remote sensing studies than the use of wavelength ranges that extend into the UV spectrum". 6. Line 113-120. Not true. There is a relatively rich literature on DOM in temperate lakes. Literature search needed. R6. Modified in text as suggested (lines 114-115): "There is a relatively rich literature on DOM in temperate lakes (e.g., Zhang et al. 2011; Read & Rose 2013; Müller et al. 2014) but few studies have focused on large shallow lakes like Lake Balaton with a strongly continental climate and hence our understanding of the variability in CDOM optical properties in these systems is comparatively poorer. 7. Lines 124-127. For objective #1, add "over the course of one year." Done For objective #3, since there were no direct measurements of the underwater light field (e.g.Kd PAR, or spectral Kd), this is not a bona fide objective. R7. We agree the fact of not providing measurements of the underwater light field makes objective 3 unreachable and this has therefore been deleted. 8. Line 174-175. One year of data is not sufficient to document a representative seasonal cycle. Caveat needed. R8. A clarification has been written specifying the seasonal variability was recorded and studied over the course of seven months (March to September 2014). 9. Line 180.

A 0.7um filter will not remove small phytoplankton, bacteria or large organic colloids. Explain why 0.2um filters were not used, and what the consequences of including small particulates might be. R9. In response to the reviewer's suggestion, the use of 0.7 um filters has been justified in the text (lines 178-183). They were selected for DOC measurements because of their compatibility with other measurements in POC (interesting in partitioning dissolved and particulate), chl- a, TSM, PC etc. Due to their larger nominal pore size, GF/F are expected to allow higher number of bacteria, viruses and colloids, which are not considered dissolved to pass though. This could lead to an overestimation of the DOC in the water samples. However, the differences due to filter are expected to be small and its importance is lessened given that DOC was only used to correlate with aCDOM. Also due to their design GF/F normally retain particles smaller then than is mentioned by the manufacture. 10. Lines 211-212. Helms et al (2008) recommended S275-295 as an indicator of in situ DOM photoprocessing for future DOM studies (not E2:E3). That needs to be acknowledged. R10. We agree with the reviewer and this text has been deleted to avoid confusion. 11. Section 2.5. CDOM photodegradation. Except for Comment #13 (below), this is a very interesting and well planned experiment. Kudos. R11. We very much appreciate the reviewer's comments on the merits of the CDOM bleaching experiment. 12. Lines 226-234. In nature, autochthonous DOM is more likely the exudates from live phytoplankton rather than dead cell remains (which would be colonized by microbes and sink out). Since exudates would be in the supernatant not the pellet, the method of grinding and digesting the pellet needs to be justified. R12. We partially agree with the reviewer, cell exudates are an important source of autochthonous CDOM but other natural processes such as grazing by zooplankton, (Levine et al. 1999)or the presence of viruses causing the lysis of phytoplankton cells (Suttle et al. 1990)can also result in release of CDOM. The process of pelleting, cleaning and breaking cells helped to mimic the effect of these processes and also ensure sufficient CDOM was produced for our experimental needs. The optical properties of the resulting CDOM were in line with expectations for autochthonous material. In the text we will comment on that fact

the physical breaking cells has the potential to release cellular material that might not be excreted the optical properties of the material produced was consistent with that for autochthonous CDOM. Moreover, this material can also be released by cell lysis due to grazing or viruses in the natural environment (Levine et al. 1999, Suttle et al. 1990) 13. Section 3.1. Seasonal variability. Caveat needed to acknowledge that seasonal changes were characterized for only 1 year, and may not represent the "average" seasonal cycle across multiple years with different weather patterns R13. As suggested by the reviewer, it has been stressed in the text that seasonal changes were only measured for one year and therefore may not represent the typical seasonal cycle observed over longer time periods. 14. Line 263. Rephrase sentence. With the exception of STO. Basin 1, the data on Fig 2 do not indicate high CDOM variability across most of the lake. In 5 of the 6 basins, DOC and a(440) were relatively stable. R14. We agree with the reviewer that this was unclear, and the sentence has been clarified. 15. Line 286. The evidence for two peaks in DOC on Fig 2 is weak at best. Rephrase. R15. We accept the reviewer's comment here and have revised the text here accordingly. There is evidence of two peaks at the inflow (summer and autumn) with the latter coinciding with the timing of peaks elsewhere. However, the evidence is weak for more than one elsewhere. 16. Line 296. Add "during July when interbasin differences were likely to be highest." R16. Accepted and modified as suggested. 17. Lines 303 to 308. This paragraph needs to be re-thought. The range of Scdom across stations is actually quite wide in the scheme of things for natural waters. Compare to Helms et al (2008) and other studies where Scdom gradients have been reported. R17. We appreciate the reviewer's comment about the significance of these results and have modified the text to better emphasise the significance of the variability observed in SCDOM. 18. Line 318. Re-think. Concluding that DOC varied more than these two optical properties seems a consequence of scaling rather than a property of the variable. Comparison of CVs or Z-scores is needed here R18. Reviewer 1 is correct and actually DOC showed greater variability through the system than for SCDOM (CV = 0.053) but smaller than for aCDOM (440) (CV = 2.065). Sentence

has been correct accordingly to this. 19. Section 3.3. Photodegradation experiment. This section needs to be re-written. The results shown on Figure 9 indicate that UV irradiation had a significant effect only on the absorbance of allochthonous cdom. This result is consistent with the data on Fig 10a. That's all that one can say with confidence about the photodegradation experiment. There is no discernible effect on Scdom or on "autochthonous" cdom in these data. R19. We agree with the reviewer and the sentence referring to autochthonous CDOM has been deleted. 20. Section 4. Discussion. This section has some overstatement, speculation and misinterpretation that needs to be removed. R20. We agree with the reviewer with some over-reaching statements and misinterpreted results and have therefore removed several statements along the section and toned down others a. L 356-363. Actually, DOM across most of the lake is relatively constant (Fig 2&5c), as observed in other large lakes. High DOM in Balaton was restricted to stations near the inflow from a high DOM river. R20A. In studies of other large lakes, such stations were likely not sampled because they skew the data and contribute little to the total lake mass of DOM. We appreciate the reviewer's comments and agree that the variability observed in the western basin was not observed over the rest of the lake. However, the influence of the DOM entering the lake from the River Zala at times stretches across the western basin, which equates to an area of approximately 100 km2. The high concentrations of DOM observed in the west are therefore not an insignificant component of the total lake DOM. Moreover, the high input of DOM will certainly influence biological processes in the highly productive western part of the lake. We have amended the text here to emphasise that marked variability was confined to the western part of the lake b. L364-374. It is an exaggeration to say that aCDOM and DOC varied seasonally throughout the system. This was only true for Basin I. In the other 4 basins, aCDOM and DOC were relatively constant throughout the year (FIG 2) R20B. The text here has been toned down in line with the reviewer's comments c. L413-424. Photomineralization would not affect Scdom, it would only affect DOM concentration. But photobleaching can have a large effect on Scdom (e.g.Helms et al, 2008), and the data for Lake Balaton shown on figs 9

and 10 suggest that the humics are the fraction of DOM that is being bleached. R20C. This sentence has been revised such that it no longer implies that mineralisation influences SCDOM and rather emphasises that the effect is on the DOM pool more broadly d. L457. Again, microbes mineralize (respire) DOM or turn it into biomass rather than bleaching it R20D. We agree with the reviewer and the text has been modified as suggested e. L471-480. This is an over-interpretation of the results shown on Fig 9. It is unsupported speculation. Omit paragraph R20D. Paragraph has been omitted as suggested f. L481-491. This is also over-interpretation and speculation. In fig 9c, there was no difference between the irradiated "autochthonous" samples and the dark controls. In 9b, there were no controls. Omit paragraph R20F. Paragraph omitted as suggested g. L492-500. The low fluorescence signal from "autochthonous" DOM is likely due to its low concentration. Re-write R20G. We agree with reviewer's 1 suggestion and the sentence has been re-written h. L 505-519. Absent measurements of the underwater light field in the lake, it is speculation to propose light limitation by DOM in such a shallow, well-mixed and eutrophic lake. Ten percent of incident solar irradiance is generally sufficient to support strong phytoplankton growth, and the authors would need to show that attenuation by DOM was sufficient to reduce downwelling light penetration below that level in the epilimnion. Re-write paragraph R20H. High DOC in the western basin certainly impacts the quality and quantity of light available for photosynthesis but the authors acknowledge we cannot demonstrate that light limitation of primary production occurs near the inflow of the River Zala from the data presented in this paper. We have revised the text to reflect this. 21. Table 1. Change notation to (mean±SD). Specify date range in heading R21. Table 1 has been modified as suggested 22. Table 2. Are these data for July 2013 or July 2014 (or both)? If both, how was interannual variation accounted for. Specify dates in heading R22. Specifications have been made on Table 2 as suggested 23. Fig 2. Check the trace for aCDOM(440). It behaves weirdly during autumn in some of the plots. Needs to be fixed R23. Graphs have been corrected as suggested 24. Fig 5. Add "measurements made in July..." to legend. R24. Changes have been made to legend

of figure 5 as suggested by Reviewer 1 25. In Fig 5c the 5 DOC categories exaggerate actual spatial variability. The three highest categories are separated by less than 1 mgC/L, while the low category spans more than 8 mg.C/L. The category indicated by a yellow X is highly questionable (zero DOC? I don't think so). Re-think this. R25. Reviewer's 1 comment made us realise some typographical error involved in the figure preparation and have been corrected 26. Fig 9. Panel 9d is superfluous. Omit R26. We agree with reviewer's comment and the panel has been deleted

Please also note the supplement to this comment:
http://www.biogeosciences-discuss.net/bg-2016-329/bg-2016-329-AC1-supplement.pdf

---

## Author Comment (AC2) · 5 Nov 2016

Response to Reviewers – Aulló-Maestro et al. Biogeosciences Discuss., doi:10.5194/bg-2016-329, 2016 Anonymous Referee #2

C1 General comments: The study has investigated the spatial and seasonal variation of chromophoric DOM in a lake at several stations leading away from an inflow from a River supplying allochthonous DOM. Various methods were used to capture change in DOM over time and spatially including analysis of the absorption coefficient spectral slope, E2:E3 ratio, SUVA and Spectral Fluorescence signals. The study shows that the

absorbance coefficient and DOC values were highest and most variable near the inflow to the lake, whereas other basins further located from the inflow had more stable values with more seasonal variability in the spectral slope of CDOM. While the study has performed what seems like a well-planned study there are some very strong statements that over reach the scope of the study. I suggest major revisions especially of the discussion.

We very much appreciate anonymous referee #2 for the time and effort put into reviewing this manuscript, his / her comments have greatly contributed to improve the manuscript.

Specific comments:

1. Line 84: What platforms are these? Please explain. R1. Here we refer to the Sentinel-2 and Sentinel-3 satellites. This can easily be clarified.

2. Line 86: Please remove "the" in the part of the sentence which reads "However, CDOM is the arguably most challenging..." R2. Modified as suggested

3. Line 118: There are quite a few studies on temperate lakes, including seasonal work, see Müller et al 2014 "Hourly, daily and seasonal variability in the absorption spectra of chromophoric dissolved...."

R3. We agree with reviewer 2 and sentence has been modified as suggested and some references added. Sentence now can be read as: "There is a relatively rich literature on DOM in temperate lakes (e.g., Zhang et al. 2011; Read & Rose 2013; Müller et al. 2014) but few studies have focused on large shallow lakes like Lake Balaton with a strongly continental climate and hence our understanding of the variability in CDOM optical properties in these systems is comparatively poorer. "

4. Line 174: Mention the time of the year samples occurred to represent seasonal variability. R4. This change has been addressed for reviewer's 1 comment and can

now be read as: "In order to capture seasonal variability in CDOM quantity and quality, water samples were collected fortnightly at 6 long-term monitoring stations on Lake Balaton over the course of seven months (March to September 2014)"

5. Line 187: Instead of referring to the summer campaign as "intensive summer campaign" change to "spatial variability" in the whole manuscript. R5. We agree with reviewer's 2 suggestion and the change has been made effective on the whole manuscript

6. Lines 190 and 193: Explain why two different instruments were used for the different campaigns. R6. The samples were analysed at different institutions with different instruments for practical reasons. This has been clarified in the text.

7. Line 194-195: It is not clear when the reference sodium azide was used. Please make this clear. R7. It was added immediately to preserve the samples; this has been clarified in the text

8. Line220: What was the temperature in the lake during this 7-day incubation? R8. The mean temperature of the lake is now stated in the methods

9. Line 211: Instead of writing "this wavelength" specify which wave length "this" refers to. R9. We agree with reviewer 2, his sentence was confusing and has therefore been deleted

10. Line 233: Was CDOM measured to know the start value? Give further explanation. R10. This data has now been added as suggested

11. Line 263: A seasonal variability in aCDOM, which was used to determine change in CDOM quantity, is not clear from figure 2. Either present statistical data backing this up or rephrase the statement. R11. We have re-written the statement as suggested

12. Line 263-269: The seasons are not shown in table 1, so the information that is referred to cannot be seen in the table. Add this information to the table. R12. We agree with reviewer's 2 observation and consider useful to include this information,

therefore, the table has been modified to indicate the month of sampling.

13. Line 270: What is the relevance of comparing August values of sCDOM with June? Do the authors mean that these are the lowest and highest values? Please make this statement clearer. R13. Yes, this is correct and is now acknowledged in the text.

14. Line292: When stating something is significantly lower the statistical data must be presented. Please add this data. R14. We no longer use the term "significant" to avoid any claim of statistical significance.

15. Line295: This sentence regarding figure 4 does not present data and should be part of the methods section. R15. This has been moved to the methods.

16. Line301: Are the values of min and max mentioned in the text also in the table? Please review. R16. They are mentioned twice and indeed replicated, the text was therefore redundant and has been deleted.

17. Line304: can this statement that there was a marked variability be made with a change of what seems to be of 0,002 on a nm scale? R17. We agree with reviewer's 2 comment and have therefore modified the sentence in the text so now it can be read: "In Kestzthely (I) basin and the western parts of Szigliget (II) basin nearest the inflow of the Zala River, higher variability was observed with lower SCDOM coefficients more than elsewhere in the lake"

18. Line 309: Where is the statistical data backing up the statement that it "varied significantly"? Please add this data. R18. The text has been modified to avoid any claim of statistical significance but ranges are reported for easy comparison to previous studies.

19. Line 310-311: Refer to table 2 for the SUVA data. R19. Modified as suggested.

20. Line 314: this statement about DOC data availability should be in methods since this cannot be seen in table 2 it is misleading to refer to it. R20. We agree with reviewer 2 and the statement has been moved to the methods section

21. Line 319: Is the correlation significant for all basins? It seems like Basin I has a strong correlation, how would it look like if they were analysed separately? R21. The relationship was only significant for the Keszthely basin. In the other basins, the variability in DOC was much lower and the sample size was small. This is now clarified in the text.

22. Line 324: where is the data for these "marked alterations"? Does this refer to aCDOM? Rewrite and connect the sentences better. R22. We agree this paragraph was confusing, it has been modified and a reference to figure 9 has been added.

23. Line 325: What was the temperature during this incubation? Can you really be sure that there was no bacterial degradation, 7 days is a long time for bacteria to degrade R23. The mean daytime lake temperature is now specified in the manuscript. We do not state there was no bacterial degradation in the treatments only that the control samples suggest bacterial degradation was minimal over the experiment and certainly a minor influence compared to UV bleaching. It possible that bacterial degradation could have been enhanced in the light but it is improbable that this would explain the differences observed between the controls and treatments. The manuscript has been revised accordingly.

DOM although you filtered through $0.2\mu$m there are always some bacteria that are small enough to get through and grow to higher abundance over time, perhaps a portion of the DOM that cannot be measured with aCDOM was taken up like what is shown in figure 10? 0.2-micron filtration typically removes >99% of bacteria from samples (Logan et al. 1993). However, it is unlikely that the filtered CDOM samples were axenic and as such it is possible that bacteria growth and metabolism of DOM in the samples contributed to the degradation of CDOM. However, comparison between the control and experimental samples clearly shows that the degradation of CDOM was greatly enhanced under solar radiation due to photobleaching

24. Line327: where the reductions statistically significant? R24. Statistical data have now been included (R2=0.952, p<0.0001)

25. Line330: Why was there an increase in the dark controls? Please discuss this. R25. As now stated in lines 493-495: "The initial decrease in slope during the early part of the experiment echoes observations by Yamashita et al. (2013) and Fichot & Benner (2012) who attributed this phenomena to microbial degradation of bioavailable CDOM (Nelson et al. 2004)"

26. Line337: When stating no "significant variation" this implies statistical significance and thus data has to be presented. Present statistical data. R26. This text has been re-written to avoid confusion.

27. Line341: same requirement as the previous comment. Show statistical data. R27. We no longer use the term "significant" to avoid any claim of statistical significance.

Discussion section:

28. Line 362: Why is it surprising that the range has not been captured in the northern latitudes? Please explain. R28. Northern boreal lakes generally have high CDOM concentrations (Curtis, 1998) and one would expect the range in these lakes to exceed that observed in Lake Balaton where catchment soils are less organic than in the peat dominated catchments of the boreal zone.

29. Lines 364-365: This statement is contradictory to your results. From figure 2 it rather seems like the aCDOM and DOC values were quite stable in most basins with variation only in Basin I at station 1 probably due to the inflow of the Zala River. Please re-write this part. R29. We agree with reviewer 2, this paragraph was confusing, it has been re-written being now: "The seasonal pattern in CDOM absorption and DOC concentration varied considerably in the western basin, but was relatively constant in other basins. The annual peak(s) in aCDOM (440) and DOC occurred in spring and/or autumn some stations (e.g., ST03, ST12, ST30) were broadly coincident with or lagged

slightly behind the annual runoff maxima suggesting a seasonal trend that was partly driven by the flushing of organic matter from catchment soils during high flow events. This pattern is common in many temperate and boreal lakes where DOC export from catchments is driven by the availability of flushable terrestrial carbon sources and the seasonality of precipitation and/or snowmelt"

30. Lines 365-367: This correlation was not shown in the results and also does not seem consistent in all basins. Where is the data for this statement? R30. This text has been re-written to avoid confusion.

31. Lines 369-373: Re-write this statement since it bases its argument on the previous statement that there could be coupling between aCDOM and DOC due to rainfall events, which was not observed in this study. R31. The peaks in aCDOM and DOC at some stations occurred in spring or autumn when runoff was high. The clear exception to this trend was the stations located near the inflow where peak aCDOM occurred in the summer due to inputs from the Kis Balaton wetland. The text has been revised to emphasise that not all stations exhibited a seasonal trend that was driven by rainfall and runoff.

32. Line 372: Isn't the Keszthely basin the same basin that is closest to the inflow of the Zala River and thus repeating what was stated in the previous sentence? R32. This has now been deleted

33. Line 385-386: Please add the reference for the water residence time. R33. Reference has been added as suggested

34. Line 395: Please add the statistical data to back-up the statement made that it was "significantly higher". R34. We no longer use the term "significant" to avoid any claim of statistical significance.

35. Lines 395-397: This information belongs in results since it is not a discussion. R35.

We agree with reviewer 2 and the sentence has been moved to results as suggested

36. Line 398: Which studies are referred to in the statement "these studies"? R36. We agree with reviewer 2 the sentence was lacking information, therefore. It was and the references of the studies added

37. Line 418: Some references needed here about photobleaching and sCDOM, this sentence seems lost here. R37. References have been added to support the statement.

38. Line 428-429: Please complete the sentence "influenced by both the provenance and subsequent transformations...." of what? R38. Further detail is now provided in the text

39. Lines 454-457: I'm not convinced that this was due to photobleaching, this section refers to figure 2, however this figure does not back-up this claim how do you rule out a dilution effect? Re-phrase. R39. We agree with reviewer 2 this sentence was misleading; the statement has been re-written

40. Line 461: Please add a reference to this paragraph. R40. References have been added to support the statement.

41. Line 465: This data needs to be compared with the control and statistical confirmation presented in the results section. R41. Comparison has been presented in the text as suggested

42. Line 481: Here if referring to allochthonous it should be less susceptible instead of more. Please change. R42. Paragraph modified

43. Line 481: There is no visible change in SCDOM in the ALLO-CDOM. Please rephrase this statement R43. We agree with the fact that there was not visible change in SCDOM for the allochthonous samples, the statement re-phrased stressing the fact that both the spectral slope and absorption coefficient for autochthonous CDOM were lower than for allochthonous samples

44. Line 481-482: Where is the statistical data to back-up the claim of statistical significance? Is this a comparison between allochthonous with autochthonous or with start values? Please add the data to the results section and re-phrase this discussion based on this. R44. Statement has been changed and as stated before, we no longer use the term "significant" to avoid any claim of statistical significance.

45. Line 492: Where is the data for fluorescence spectra of autochthonous material? Figure 10a an10b only present allochthonous R45. There were more than ten orders of magnitude difference in fluorescence intensity between CDOM allo and CDOM auto samples, presumably driven by the difference in concentration. Given the low concentrations of CDOM, after Milli-Q correction, there was no measurable fluorescence signal for the autochthonous samples. Therefore, fluorescence spectra of autochthonous material have not been presented in figure 10

46. Line 495: where is this data? R46. Please refer to response above.

47. Line 500: could this loss not be due to bacterial degradation? R47. Indeed, we suggest that this loss is due to photobleaching and not to bacterial degradation. Statement stressed in the text to avoid confusion.

48. Line 505-506: Please add a reference to this sentence. R48. References have been added to support the statement.

49. Line 506: what is meant by "elsewhere"? R49. Paragraph has been re-written

50. Line 505-509: This is a very strong statement that cannot be proven with the data from this study. Please re-write. R50. We agree and have toned down the statement in line with the reviewer's comments

51. Line 512: Also this statement is too bold since this was not within the scope of this study. R51. Statement deleted

52. Line 522-524: Please add a reference to this statement. R52. Statement deleted

53. Line 547: Isn't the contribution of wetlands well known? Remove "novel". R53. Removed as suggested

Technical comments:

54. Line 70-71: Please review this sentence, it seems like information is being repeated and there is a misuse of the word "whilst". R54. Sentence has been re-written as suggested by reviewer's 2

55. Line 71: In the same sentence as the above comment "...this fulfilling important role..." should probably be "thus fulfilling an important role". R55. Sentence re-written as suggested by reviewer's 2

56. Line 75: can CDOM have a behaviour? Perhaps property could be used instead. R56. Sentence re-written as suggested by reviewer's 2

57. Line 87: should be changed to "for reliable estimation of remotely..." Please change. R57. Sentence re-written as suggested by reviewer's 2

58. Line 89: should be changed to "studies have explored the application..." R58. Sentence re-written as suggested by reviewer's 2

59. Line 97: change to "size of DOM molecules..." R59. Sentence re-written as suggested by reviewer's 2

60. Line 98: I think the authors mean larger/greater molecules, not higher. R60. This text has been re-written to avoid confusion.

61. Lines 131-133: Please add references to this information about the study area. R61. References have been added as suggested

62. Line 136: should be changed to "...at that time of the year..." R62. Sentence has been changed as suggested

63. Lines 162-164: Please add a reference to this statement. R63. References have been added as suggested

64. Line 165: what is meant by "...less noticeable..."? Less than what? R64. This text has been re-written to avoid confusion.

65. Line 219: I suggest moving "fifty-six" to Line 222 so it reads "Fifty-six samples were taken in total of which 21 were composed of..." R65. Sentence changed as suggested

66. Line 228: Please add a reference to the dominance of the phytoplankton in this particular lake. R66. References have been added as suggested

67. Line311: if reference to figure 6d and 6e is made then SUVA should be mentioned first and then E2/E3 ratio to be consistent. Then you can say that it refers to those figures respectively. R67. Sentence modified as suggested

68. Line315: mean value in table 2 for Keszthely basin is 9.66 not 9.67 as it says in the text. Which is correct? Please review. R68. Sentence modified to 9.66, corrected values

69. Line317: Do you mean with increasing distance from Zala River? Sentence modified to avoid confusion R69. Yes, DOC concentrations slowly decreased with increasing distance from Zala River

70. Line 317: remove "in" before the word similarly. R70. Sentence changed as suggested

71. Line346: Change to "there were more than ten orders..." R71. Sentence changed as suggested

Reference list: I have not checked the reference list.

72. Line434: How does this statement connect with the data in this study: "previous studies have also found marked differences in the E2:E3 between natural waters..." Present the data from the study and then connect with what other studies have found.

[Figure]

R72. Data from the study is shown in results, lines 320-321 and Table 3. Statement modified to avoid confusion. References added to support the statement.

73. Line446: Remove "in" after Lake Balaton. R73. Sentence changed as suggested

74. Line450: change "sensitive" to sensitivity. R74. Changed as suggested

75. Line456: Add: and, between the two ranges. R75. Changed as suggested

76. Line530: change to "new approaches are needed..." R76. Sentence modified as suggested

77. Tables 1 and 2: Is there a reason why values are stated as Max-Min instead of Min-Max? Consider changing to better fit with standard way of reporting such values. R77. Modified as suggested

78. Figure2: the lines connecting data points for aCDOM seem to connect in a strange way or to be disconnected. Please review and fix. Figure 9: add to legend what "DC" refers to, dark control? R78. Figure modified as suggested

Please also note the supplement to this comment:
http://www.biogeosciences-discuss.net/bg-2016-329/bg-2016-329-AC2-supplement.pdf

---

## Author Response (AR1)

Dear Editor,

We thank you and the reviewers for your careful reading of our manuscript entitled "Spatio-seasonal variability of chromophoric dissolved organic matter absorption and responses to photobleaching in a large shallow temperate lake". Here, we submit the revised version of it for consideration to publication. The manuscript has been prepared in accordance with the Instructions for Authors and none of the authors have any conflicts of interest.

Following we will address all referee's comments organized such that first the reviewer comments are given in regular font, directly followed by our response in red and in italics. Finally, you can find a list of the most relevant changes made in the manuscript, and a marked-up version of it.

**Response to Reviewer 1 – Aulló-Maestro et al. Biogeosciences Discuss., doi:10.5194/bg-2016-329, 2016**

Reviewer comments appear as normal text

*Our responses appear in red and italicised*

General comments. This paper documents variability in DOM along a longitudinal transect in Lake Balaton during 2014. The authors report that DOM concentrations generally ranged from 8 to 16 mgC/L in surface waters (which is relatively high), with values above 10 mgC/L mostly in the eastern end of the lake which receives inflow from a large wetland. They also report four optical properties of the DOM (absorbance, spectral slope coefficient, SUVA and E2/E3) as potential indices of DOM source or internal processing (especially photodegradation). Unfortunately, the choice of wavelengths for some of the optical measurements was not optimal, thus limiting comparison with other systems and studies.

*We agree with the reviewer that this can limit the comparison with other systems and studies, however, and as justified after in the text, this paper is strongly motivated by the understanding of the changes on the inherent optical properties of CDOM particularly with reference to the implications for remote sensing. In this framework, remote sensing studies often use 412 nm or 440 nm (Carder et al. 1989; Nelson et al. 1998; Schwarz et al. 2002) to describe CDOM absorption because information in the UV cannot be obtained from space. For this reason, and to ensure consistency with previous publications, our results are more strongly based on absorption in the blue rather than the UV*

Although the paper reports apparently new information about Lake Balaton, there are several matters in the text, tables and figures that will require re-thinking and major revision.

*We wish to thank the reviewer for this very thorough and constructive review. We agree with most of the suggestions and have therefore modified the text according to them. We think they have greatly enhanced the quality of the paper*

Specific comments.

1. Abstract. Line 35. The data on Fig 9c and 9d are not convincing evidence that UV irradiation caused any change in the absorbance of "autochthonous" DOM. Omit sentence

*The sentence has been omitted as suggested*

2. Abstract. Omit last two sentences. Nothing new here.

*The sentence has been omitted as suggested*

3. Lines 70-73. Unclear sentence. What nutrients are we talking about? Re-think and re-write.

*We agree it sounded like an incomplete sentence and have therefore rephrased it to now read as:*

*"CDOM can be remineralised by bacteria acting as a source of inorganic nutrients (Buchan et al. 2014), which is important for phytoplankton nutrition"*

4. Line 93. The wavelength 440nm is not commonly used for CDOM. 440nm is routinely used for Chl-a because it is the absorbance maximum for that pigment. The preferred wavelength for CDOM is in the UV, usually less than 370nm. Choice of 440 (blue) rather than UV needs to be justified.

*As justified later in the text (lines 196–198) and explained before, the fact of this study being strongly motivated by the understanding of the changes on the inherent optical properties of CDOM particularly with reference to the implications for remote sensing and to ensure consistency with previous publications, our results are based on CDOM absorption measurements at 440 nm*

5. Line 95. The spectral slope coefficient recommended by Helms et al (2008) and by Fichot and Benner (2012) is calculated over the UV range 275-295 nm. But in Balaton, the authors used a very different range (350-500 nm). Choice of this range needs to be justified given that the authors cite Helms et al and Fichot and Benner.

*We agree our choice needs to be justified and as modified in the text (line 201-203): "This range of calculation was consistent with Babin et al. 2003 and Matsuoka et al. 2012 amongst others and is more relevant to remote sensing studies than the use of wavelength ranges that extend into the UV spectrum"*

6. Line 113-120. Not true. There is a relatively rich literature on DOM in temperate lakes. Literature search needed.

*Modified in text as suggested (lines 114-115): "There is a relatively rich literature on DOM in temperate lakes (e.g., Zhang et al. 2011; Read & Rose 2013; Müller et al. 2014) but few studies have focused on large shallow lakes like Lake Balaton with a strongly continental climate and hence our understanding of the variability in CDOM optical properties in these systems is comparatively poorer*

7. Lines 124-127. For objective #1, add "over the course of one year." Done For objective #3, since there were no direct measurements of the underwater light field (e.g.Kd PAR, or spectral Kd), this is not a bona fide objective.

*We agree the fact of not providing measurements of the underwater light field makes objective 3 unreachable and this has therefore been deleted*

8. Line 174-175. One year of data is not sufficient to document a representative seasonal cycle. Caveat needed.

*A clarification has been written specifying the seasonal variability was recorded and studied over the course of seven months (March to September 2014)*

9. Line 180. A 0.7um filter will not remove small phytoplankton, bacteria or large organic colloids. Explain why 0.2um filters were not used, and what the consequences of including small particulates might be.

*In response to the reviewer's suggestion, the use of 0.7 um filters has been justified in the text (lines 178-183). They were selected for DOC measurements because of their compatibility with other measurements in POC (interesting in partitioning dissolved and particulate), chl- a, TSM, PC etc. Due to their larger nominal pore size, GF/F are expected to allow higher number of bacteria, viruses and colloids, which are not considered dissolved to pass though. This could lead to an overestimation of the DOC in the water samples. However, the differences due to filter are expected to be small and its importance is lessened given that DOC was only used to correlate with $a_{CDOM}$. Also due to their design GF/F normally retain particles smaller then than is mentioned by the manufacture*

10. Lines 211-212. Helms et al (2008) recommended S275-295 as an indicator of in situ DOM photoprocessing for future DOM studies (not E2:E3). That needs to be acknowledged.

*We agree with the reviewer and this text has been deleted to avoid confusion*

11. Section 2.5. CDOM photodegradation. Except for Comment #13 (below), this is a very interesting and well planned experiment. Kudos.

*We very much appreciate the reviewer's comments on the merits of the CDOM bleaching experiment*

12. Lines 226-234. In nature, autochthonous DOM is more likely the exudates from live phytoplankton rather than dead cell remains (which would be colonized by microbes and sink out). Since exudates would be in the supernatant not the pellet, the method of grinding and digesting the pellet needs to be justified.

*We partially agree with the reviewer, cell exudates are an important source of autochthonous CDOM but other natural processes such as grazing by zooplankton,* (Levine et al. 1999)*or the presence of viruses causing the lysis of phytoplankton cells* (Suttle et al. 1990)*can also result*

*in release of CDOM. The process of pelleting, cleaning and breaking cells helped to mimic the effect of these processes and also ensure sufficient CDOM was produced for our experimental needs. The optical properties of the resulting CDOM were in line with expectations for autochthonous material*

*In the text we will comment on that fact the physical breaking cells has the potential to release cellular material that might not be excreted the optical properties of the material produced was consistent with that for autochthonous CDOM. Moreover, this material can also be released by cell lysis due to grazing or viruses in the natural environment (Levine et al. 1999, Suttle et al. 1990)*

13. Section 3.1. Seasonal variability. Caveat needed to acknowledge that seasonal changes were characterized for only 1 year, and may not represent the "average" seasonal cycle across multiple years with different weather patterns

*As suggested by the reviewer, it has been stressed in the text that seasonal changes were only measured for one year and therefore may not represent the typical seasonal cycle observed over longer time period*

14. Line 263. Rephrase sentence. With the exception of STO. Basin 1, the data on Fig 2 do not indicate high CDOM variability across most of the lake. In 5 of the 6 basins, DOC and a(440) were relatively stable.

*We agree with the reviewer that this was unclear, and the sentence has been clarified*

15. Line 286. The evidence for two peaks in DOC on Fig 2 is weak at best. Rephrase.

*We accept the reviewer's comment here and have revised the text here accordingly. There is evidence of two peaks at the inflow (summer and autumn) with the latter coinciding with the timing of peaks elsewhere. However, the evidence is weak for more than one elsewhere*

16. Line 296. Add "during July when interbasin differences were likely to be highest."

*Accepted and modified as suggested*

17. Lines 303 to 308. This paragraph needs to be re-thought. The range of Scdom across stations is actually quite wide in the scheme of things for natural waters. Compare to Helms et al (2008) and other studies where Scdom gradients have been reported.

*We appreciate the reviewer's comment about the significance of these results and have modified the text to better emphasise the significance of the variability observed in $S_{CDOM}$*

18. Line 318. Re-think. Concluding that DOC varied more than these two optical properties seems a consequence of scaling rather than a property of the variable. Comparison of CVs or Z-scores is needed here

*Reviewer 1 is correct and actually DOC showed greater variability through the system than for $S_{CDOM}$ (CV = 0.053) but smaller than for $a_{CDOM}$ (440) (CV = 2.065). Sentence has been correct accordingly to this*

19. Section 3.3. Photodegradation experiment. This section needs to be re-written. The results shown on Figure 9 indicate that UV irradiation had a significant effect only on the absorbance of allochthonous cdom. This result is consistent with the data on Fig 10a. That's all that one can say with confidence about the photodegradation experiment. There is no discernible effect on Scdom or on "autochthonous" cdom in these data.

*We agree with the reviewer and the sentence referring to autochthonous CDOM has been deleted*

20. Section 4. Discussion. This section has some overstatement, speculation and misinterpretation that needs to be removed.

*We agree with the reviewer with some over-reaching statements and misinterpreted results and have therefore removed several statements along the section and toned down others*

   a. L 356-363. Actually, DOM across most of the lake is relatively constant (Fig 2&5c), as observed in other large lakes. High DOM in Balaton was restricted to stations near the inflow from a high DOM river. In studies of other large lakes, such stations were likely not sampled because they skew the data and contribute little to the total lake mass of DOM.

*We appreciate the reviewer's comments and agree that the variability observed in the western basin was not observed over the rest of the lake.  However, the influence of the DOM entering the lake from the River Zala at times stretches across the western basin, which equates to an area of approximately 100 km2. The high concentrations of DOM observed in the west are therefore not an insignificant component of the total lake DOM.  Moreover, the high input of DOM will certainly influence biological processes in the highly productive western part of the lake. We have amended the text here to emphasise that marked variability was confined to the western part of the lake*

   b. L364-374. It is an exaggeration to say that aCDOM and DOC varied seasonally throughout the system. This was only true for Basin I. In the other 4 basins, aCDOM and DOC were relatively constant throughout the year (FIG 2)

*The text here has been toned down in line with the reviewer's comments*

c. L413-424. Photomineralization would not affect Scdom, it would only affect DOM concentration. But photobleaching can have a large effect on Scdom (e.g.Helms et al, 2008), and the data for Lake Balaton shown on figs 9 and 10 suggest that the humics are the fraction of DOM that is being bleached.

*This sentence has been revised such that it no longer implies that mineralisation influences SCDOM and rather emphasises that the effect is on the DOM pool more broadly*

d. L457. Again, microbes mineralize (respire) DOM or turn it into biomass rather than bleaching it

*We agree with the reviewer and the text has been modified as suggested*

e. L471-480. This is an over-interpretation of the results shown on Fig 9. It is unsupported speculation. Omit paragraph

*Paragraph has been omitted as suggested*

f. L481-491. This is also over-interpretation and speculation. In fig 9c, there was no difference between the irradiated "autochthonous" samples and the dark controls. In 9b, there were no controls. Omit paragraph

*Paragraph omitted as suggested*

g. L492-500. The low fluorescence signal from "autochthonous" DOM is likely due to its low concentration. Re-write

*We agree with reviewer's 1 suggestion and the sentence has been re-written*

h. L 505-519. Absent measurements of the underwater light field in the lake, it is speculation to propose light limitation by DOM in such a shallow, well-mixed and eutrophic lake. Ten percent of incident solar irradiance is generally sufficient to support strong phytoplankton growth, and the authors would need to show that attenuation by DOM was sufficient to reduce downwelling light penetration below that level in the epilimnion. Re-write paragraph

*High DOC in the western basin certainly impacts the quality and quantity of light available for photosynthesis but the authors acknowledge we cannot demonstrate that light limitation of primary production occurs near the inflow of the River Zala from the data presented in this paper. We have revised the text to reflect this*

21. Table 1. Change notation to (mean±SD). Specify date range in heading

*Table 1 has been modified as suggested*

22. Table 2. Are these data for July 2013 or July 2014 (or both)? If both, how was interannual variation accounted for. Specify dates in heading

*Specifications have been made on Table 2 as suggested*

23. Fig 2. Check the trace for $a_{CDOM}$ (440). It behaves weirdly during autumn in some of the plots. Needs to be fixed

*Graphs have been corrected as suggested*

24. Fig 5. Add "measurements made in July…" to legend.

*Changes have been made to legend of figure 5 as suggested by Reviewer 1*

25. In Fig 5c the 5 DOC categories exaggerate actual spatial variability. The three highest categories are separated by less than 1 mgC/L, while the low category spans more than 8 mg.C/L. The category indicated by a yellow X is highly questionable (zero DOC? I don't think so). Re-think this.

*Reviewer's 1 comment made us realise some typographical error involved in the figure preparation and have been corrected*

26. Fig 9. Panel 9d is superfluous. Omit

*We agree with reviewer's comment and the panel has been deleted*

**Response to Reviewers – Aulló-Maestro et al. Biogeosciences Discuss., doi:10.5194/bg-2016-329, 2016**

**Anonymous Referee #2**

Reviewer comments appear as normal text

*Our responses appear in red and italicised*

General comments: The study has investigated the spatial and seasonal variation of chromophoric DOM in a lake at several stations leading away from an inflow from a River supplying allochthonous DOM. Various methods were used to capture change in DOM over time and spatially including analysis of the absorption coefficient spectral slope, E2:E3 ratio, SUVA and Spectral Fluorescence signals. The study shows that the absorbance coefficient and DOC values were highest and most variable near the inflow to the lake, whereas other basins further located from the inflow had more stable values with more seasonal variability in the spectral slope of CDOM. While the study has performed what seems like a well-planned study there are some very strong statements that over reach the scope of the study. I suggest major revisions especially of the discussion.

*We very much appreciate anonymous referee #2 for the time and effort put into reviewing this manuscript, his / her comments have greatly contributed to improve the manuscript*

Specific comments:

1. Line 84: What platforms are these? Please explain.

*Here we refer to the Sentinel-2 and Sentinel-3 satellites.  This can easily be clarified*

2. Line 86: Please remove "the" in the part of the sentence which reads "However, CDOM is the arguably most challenging..."

*Modified as suggested*

3. Line 118: There are quite a few studies on temperate lakes, including seasonal work, see Müller et al 2014 "Hourly, daily and seasonal variability in the absorption spectra of chromophoric dissolved...."

*We agree with reviewer 2 and sentence has been modified as suggested and some references added. Sentence now can be read as: "There is a relatively rich literature on DOM in temperate lakes (e.g., Zhang et al. 2011; Read & Rose 2013; Müller et al. 2014) but few studies have focused on large shallow*

*lakes like Lake Balaton with a strongly continental climate and hence our understanding of the variability in CDOM optical properties in these systems is comparatively poorer. "*

4. Line 174: Mention the time of the year samples occurred to represent seasonal variability.

*This change has been addressed for reviewer's 1 comment and can now be read as: "In order to capture seasonal variability in CDOM quantity and quality, water samples were collected fortnightly at 6 long-term monitoring stations on Lake Balaton over the course of seven months (March to September 2014) "*

5. Line 187: Instead of referring to the summer campaign as "intensive summer campaign" change to "spatial variability" in the whole manuscript.

*We agree with reviewer's 2 suggestion and the change has been made effective on the whole manuscript*

6. Lines 190 and 193: Explain why two different instruments were used for the different campaigns.

*The samples were analysed at different institutions with different instruments for practical reasons. This has been clarified in the text*

7. Line 194-195: It is not clear when the reference sodium azide was used. Please make this clear.

*It was added immediately to preserve the samples; this has been clarified in the text*

8. Line220: What was the temperature in the lake during this 7-day incubation?

*The mean temperature of the lake is now stated in the methods*

9. Line 211: Instead of writing "this wavelength" specify which wave length "this" refers to.

*We agree with reviewer 2, his sentence was confusing and has therefore been deleted*

10. Line 233: Was CDOM measured to know the start value? Give further explanation.

*This data has now been added as suggested*

11. Line 263: A seasonal variability in aCDOM, which was used to determine change in CDOM quantity, is not clear from figure 2. Either present statistical data backing this up or rephrase the statement.

*We have re-written the statement as suggested*

12. Line 263-269: The seasons are not shown in table 1, so the information that is referred to cannot be seen in the table. Add this information to the table.

*We agree with reviewer's 2 observation and consider useful to include this information, therefore, the table has been modified to indicate the month of sampling*

13. Line 270: What is the relevance of comparing August values of sCDOM with June? Do the authors mean that these are the lowest and highest values? Please make this statement clearer.

*Yes, this is correct and is now acknowledged in the text*

14. Line 292: When stating something is significantly lower the statistical data must be presented. Please add this data.

*Values of significance have been added*

15. Line 295: This sentence regarding figure 4 does not present data and should be part of the methods section.

*This has been moved to the methods*

16. Line 301: Are the values of min and max mentioned in the text also in the table? Please review.

*They are mentioned twice and indeed replicated, the text was therefore redundant and has been deleted*

17. Line 304: can this statement that there was a marked variability be made with a change of what seems to be of 0,002 on a nm scale?

*We agree with reviewer's 2 comment and have therefore modified the sentence in the text so now it can be read: "In Keszthely (I) basin and the western parts of Szigliget (II) basin nearest the inflow of the Zala River, higher variability was observed with lower SCDOM coefficients more than elsewhere in the lake"*

18. Line 309: Where is the statistical data backing up the statement that it "varied significantly"? Please add this data.

*The text has been modified to avoid any claim of statistical significance but ranges are reported for easy comparison to previous studies*

19. Line 310-311: Refer to table 2 for the SUVA data.

*Modified as suggested*

20. Line 314: this statement about DOC data availability should be in methods since this cannot be seen in table 2 it is misleading to refer to it.

*We agree with reviewer 2 and the statement has been moved to the methods section*

21. Line 319: Is the correlation significant for all basins? It seems like Basin I has a strong correlation, how would it look like if they were analysed separately?

*The relationship was only significant for the Keszthely basin. In the other basins, the variability in DOC was much lower and the sample size was small. This is now clarified in the text*

22. Line 324: where is the data for these "marked alterations"? Does this refer to aCDOM? Rewrite and connect the sentences better.

*We agree this paragraph was confusing, it has been modified and a reference to figure 9 has been added*

23. Line 325: What was the temperature during this incubation? Can you really be sure that there was no bacterial degradation, 7 days is a long time for bacteria to degrade

*The mean daytime lake temperature is now specified in the manuscript. We do not state there was no bacterial degradation in the treatments only that the control samples suggest bacterial degradation was minimal over the experiment and certainly a minor influence compared to UV bleaching. It possible that bacterial degradation could have been enhanced in the light but it is improbable that this would explain the differences observed between the controls and treatments. The manuscript has been revised accordingly*

DOM although you filtered through 0.2µm there are always some bacteria that are small enough to get through and grow to higher abundance over time, perhaps a portion of the DOM that cannot be measured with aCDOM was taken up like what is shown in figure 10?

*0.2-micron filtration typically removes >99% of bacteria from samples (Logan et al. 1993). However, it is unlikely that the filtered CDOM samples were axenic and as such it is possible that bacteria growth and metabolism of DOM in the samples contributed to the degradation of CDOM. However, comparison between the control and experimental samples clearly shows that the degradation of CDOM was greatly enhanced under solar radiation due to photobleaching*

24. Line327: where the reductions statistically significant?

*Statistical data have now been included (ANOVA, $p<0.01$)*

25. Line330: Why was there an increase in the dark controls? Please discuss this.

*As now stated in lines 493-495: "The initial decrease in slope during the early part of the experiment echoes observations by Yamashita et al. (2013) and Fichot & Benner (2012) who*

*attributed this phenomena to microbial degradation of bioavailable CDOM (Nelson et al. 2004)"*

26.Line337: When stating no "significant variation" this implies statistical significance and thus data has to be presented. Present statistical data.

*This text has been re-written to avoid confusion*

27.Line341: same requirement as the previous comment. Show statistical data.

*Data added as requested*

Discussion section:

28.Line 362: Why is it surprising that the range has not been captured in the northern latitudes? Please explain.

*Northern boreal lakes generally have high CDOM concentrations (Curtis, 1998) and one would expect the range in these lakes to exceed that observed in Lake Balaton where catchment soils are less organic than in the peat dominated catchments of the boreal zone*

29.Lines 364-365: This statement is contradictory to your results. From figure 2 it rather seems like the aCDOM and DOC values were quite stable in most basins with variation only in Basin I at station 1 probably due to the inflow of the Zala River. Please re-write this part.

*We agree with reviewer 2, this paragraph was confusing, it has been re-written being now: "The seasonal pattern in CDOM absorption and DOC concentration varied considerably in the western basin, but was relatively constant in other basins. The annual peak(s) in aCDOM (440) and DOC occurred in spring and/or autumn some stations (e.g., ST03, ST12, ST30) were broadly coincident with or lagged slightly behind the annual runoff maxima suggesting a seasonal trend that was partly driven by the flushing of organic matter from catchment soils during high flow events. This pattern is common in many temperate and boreal lakes where DOC export from catchments is driven by the availability of flushable terrestrial carbon sources and the seasonality of precipitation and/or snowmelt"*

30.Lines 365-367: This correlation was not shown in the results and also does not seem consistent in all basins. Where is the data for this statement?

*This text has been re-written to avoid confusion*

31. Lines 369-373: Re-write this statement since it bases its argument on the previous statement that there could be coupling between aCDOM and DOC due to rainfall events, which was not observed in this study.

*The peaks in aCDOM and DOC at some stations occurred in spring or autumn when runoff was high. The clear exception to this trend was the stations located near the inflow where peak aCDOM occurred in the summer due to inputs from the Kis Balaton wetland. The text has been revised to emphasise that not all stations exhibited a seasonal trend that was driven by rainfall and runoff*

32. Line 372: Isn't the Keszthely basin the same basin that is closest to the inflow of the Zala River and thus repeating what was stated in the previous sentence?

*This has now been deleted*

33. Line 385-386: Please add the reference for the water residence time.

*Reference has been added as suggested*

34. Line 395: Please add the statistical data to back-up the statement made that it was "significantly higher".

*Statistical data has been added as requested*

35. Lines 395-397: This information belongs in results since it is not a discussion.

*We agree with reviewer 2 and the sentence has been moved to results as suggested*

36. Line 398: Which studies are referred to in the statement "these studies"?

*We agree with reviewer 2 the sentence was lacking information, therefore. It was and the references of the studies added*

37. Line 418: Some references needed here about photobleaching and sCDOM, this sentence seems lost here.

*References have been added to support the statement*

38. Line 428-429: Please complete the sentence "influenced by both the provenance and subsequent transformations...." of what?

*Further detail is now provided in the text*

39. Lines 454-457: I'm not convinced that this was due to photobleaching, this section refers to figure 2, however this figure does not back-up this claim how do you rule out a dilution effect? Re-phrase.

*We agree with reviewer 2 this sentence was misleading; the statement has been re-written*

40. Line 461: Please add a reference to this paragraph.

*References have been added to support the statement*

41. Line 465: This data needs to be compared with the control and statistical confirmation presented in the results section.

*Comparison has been presented in the text as suggested*

42. Line 481: Here if referring to allochthonous it should be less susceptible instead of more. Please change.

*Paragraph modified*

43. Line 481: There is no visible change in SCDOM in the ALLO-CDOM. Please rephrase this statement

*We agree with the fact that there was not visible change in SCDOM for the allochthonous samples, the statement re-phrased stressing the fact that both the spectral slope and absorption coefficient for autochthonous CDOM were lower than for allochthonous samples*

44. Line 481-482: Where is the statistical data to back-up the claim of statistical significance? Is this a comparison between allochthonous with autochthonous or with start values? Please add the data to the results section and re-phrase this discussion based on this.

*Data added as requested*

45. Line 492: Where is the data for fluorescence spectra of autochthonous material? Figure 10a an10b only present allochthonous

*There were more than ten orders of magnitude difference in fluorescence intensity between CDOM allo and CDOM auto samples, presumably driven by the difference in concentration. Given the low concentrations of CDOM, after Milli-Q correction, there was no measurable fluorescence signal for the autochthonous samples.  Therefore, fluorescence spectra of autochthonous material have not been presented in figure 10*

46. Line 495: where is this data?

*Please refer to response above*

47. Line 500: could this loss not be due to bacterial degradation?

*Indeed, we suggest that this loss is due to photobleaching and not to bacterial degradation. Statement stressed in the text to avoid confusion*

48. Line 505-506: Please add a reference to this sentence.

*References have been added to support the statement*

49. Line 506: what is meant by "elsewhere"?

*Paragraph has been re-written*

50. Line 505-509: This is a very strong statement that cannot be proven with the data from this study. Please re-write.

*We agree and have toned down the statement in line with the reviewer's comments*

51. Line 512: Also this statement is too bold since this was not within the scope of this study.

*Statement deleted*

52. Line 522-524: Please add a reference to this statement.

*Statement deleted*

53. Line 547: Isn't the contribution of wetlands well known? Remove "novel".

*Removed as suggested*

Technical comments:

54. Line 70-71: Please review this sentence, it seems like information is being repeated and there is a misuse of the word "whilst".

*Sentence has been re-written as suggested by reviewer's 2*

55. Line 71: In the same sentence as the above comment "...this fulfilling important role..." should probably be "thus fulfilling an important role".

*Sentence re-written as suggested by reviewer's 2*

56. Line 75: can CDOM have a behaviour? Perhaps property could be used instead.

*Sentence re-written as suggested by reviewer's 2*

57. Line 87: should be changed to "for reliable estimation of remotely..." Please change.

*Sentence re-written as suggested by reviewer's 2*

58. Line 89: should be changed to "studies have explored the application..."

*Sentence re-written as suggested by reviewer's 2*

59. Line 97: change to "size of DOM molecules..."

*Sentence re-written as suggested by reviewer's 2*

60. Line 98: I think the authors mean larger/greater molecules, not higher.

*This text has been re-written to avoid confusion.*

61. Lines 131-133: Please add references to this information about the study area.

*References have been added as suggested*

62. Line 136: should be changed to "...at that time of the year..."

*Sentence has been changed as suggested*

63. Lines 162-164: Please add a reference to this statement.

*References have been added as suggested*

64. Line 165: what is meant by "...less noticeable..."? Less than what?

*This text has been re-written to avoid confusion*

65. Line 219: I suggest moving "fifty-six" to Line 222 so it reads "Fifty-six samples were taken in total of which 21 were composed of..."

*Sentence changed as suggested*

66. Line 228: Please add a reference to the dominance of the phytoplankton in this particular lake.

*References have been added as suggested*

67. Line311: if reference to figure 6d and 6e is made then SUVA should be mentioned first and then E2/E3 ratio to be consistent. Then you can say that it refers to those figures respectively.

*Sentence modified as suggested*

68. Line315: mean value in table 2 for Keszthely basin is 9.66 not 9.67 as it says in the text. Which is correct? Please review.

*Sentence modified to 9.66, corrected values*

69. Line317: Do you mean with increasing distance from Zala River?

*Sentence modified to avoid confusion*

*Yes, DOC concentrations slowly decreased with increasing distance from Zala River*

70. Line 317: remove "in" before the word similarly.

*Sentence changed as suggested*

71. Line346: Change to "there were more than ten orders..."

*Sentence changed as suggested*

**Reference list:** I have not checked the reference list.

72. Line434: How does this statement connect with the data in this study: "previous studies have also found marked differences in the E2:E3 between natural waters..." Present the data from the study and then connect with what other studies have found.

*Data from the study is shown in results, lines 320-321 and Table 3. Statement modified to avoid confusion. References added to support the statement*

73. Line446: Remove "in" after Lake Balaton.

*Sentence changed as suggested*

74. Line450: change "sensitive" to sensitivity.

*Changed as suggested*

75. Line456: Add: and, between the two ranges.

*Changed as suggested*

76. Line530: change to "new approaches are needed..."

*Sentence modified as suggested*

77. Tables 1 and 2: Is there a reason why values are stated as Max-Min instead of Min-Max? Consider changing to better fit with standard way of reporting such values.

*Modified as suggested*

78. Figure2: the lines connecting data points for aCDOM seem to connect in a strange way or to be disconnected. Please review and fix. Figure 9: add to legend what "DC" refers to, dark control?

*Figure modified as suggested*

**List of relevant changes in the manuscript**

1. Improvements in grammar and writing as mentioned by the referees
2. Re-thought and re-written some statements as suggested
3. Addition of more statistics to support some statements
4. Modification and addition of tables and figures as suggested
5. Addition of some citations as suggested

[revised manuscript text omitted]